# Sticks and Stones, a conserved cell surface ligand for the Type IIa RPTP Lar, regulates neural circuit wiring in *Drosophila*

**Namrata Bali\*, Hyung-Kook (Peter) Lee, Kai Zinn\***

Division of Biology and Biological Engineering, California Institute of Technology, Pasadena, United States

**Abstract** Type IIa receptor-like protein tyrosine phosphatases (RPTPs) are essential for neural development. They have cell adhesion molecule (CAM)-like extracellular domains that interact with cell-surface ligands and coreceptors. We identified the immunoglobulin superfamily CAM Sticks and Stones (Sns) as a new partner for the *Drosophila* Type IIa RPTP Lar. Lar and Sns bind to each other in embryos and in vitro, and the human Sns ortholog, Nephrin, binds to human Type IIa RPTPs. Genetic analysis shows that Lar and Sns function together to regulate larval neuromuscular junction development, axon guidance in the mushroom body (MB), and innervation of the optic lobe (OL) medulla by R7 photoreceptors. In the neuromuscular system, Lar and Sns are both required in motor neurons, and may function as coreceptors. In the MB and OL, however, the relevant Lar-Sns interactions are in *trans* (between neurons), so Sns functions as a Lar ligand in these systems.

**\*For correspondence:**
nbali@caltech.edu (NB);
zinnk@caltech.edu (KZ)

**Competing interest:** The authors declare that no competing interests exist.

## Editor's evaluation

This article claims to identify a long-sought ligand for the receptor protein tyrosine phosphatase Lar that mediates its functions in neuromuscular junction development, mushroom body development, and photoreceptor axon targeting. This would be of interest to many developmental neurobiologists.

## Introduction

Neural circuit assembly involves axon pathfinding, target selection, and establishment of synaptic connections with appropriate targets. Cell adhesion molecules (CAMs) play important roles in all of these processes. CAMs can initiate cell–cell contact and recruit pre- and postsynaptic proteins to direct synapse formation and maturation. They usually have an extracellular domain (ECD) that interacts with other CAMs, either homophilically or heterophilically, a transmembrane domain, and an intracellular domain that transduces signals.

Receptor-like protein tyrosine phosphatases (RPTPs) are transmembrane signaling receptors with CAM-like ECDs and cytoplasmic regions containing one or two PTP domains. *Drosophila* has six RPTPs, four of which are primarily expressed in the nervous system. Type IIa RPTPs (also known as LAR or R2A RPTPs) have large ECDs containing immunoglobulin superfamily (IgSF) domains and fibronectin type III (FNIII) repeats. They bind heterophilically to ligands and coreceptors (reviewed by *Coles et al., 2015*; *Fukai and Yoshida, 2021*). *Drosophila* Lar is a Type IIa RPTP with IgSF and FNIII domains that is orthologous to three mammalian Type IIa RPTPs: PTPRF (LAR), PTPRD (PTPδ, R-PTP-δ), and PTPRS (PTPσ, R-PTP-σ). *Caenorhabditis elegans* has a single Type IIa RPTP, PTP-3.

Lar is selectively expressed in neurons during development. It regulates motor axon guidance in embryos and determines the numbers of synaptic boutons in neuromuscular junctions (NMJs) in larvae (*Desai et al., 1997*; *Kaufmann et al., 2002*; *Krueger et al., 1996*). Lar is required for R7 photoreceptor axon targeting in the optic lobe (OL) (*Clandinin et al., 2001*; *Maurel-Zaffran et al., 2001*) and for development of the lobes of the larval mushroom body (MB) (*Kurusu and Zinn, 2008*).

The heparan sulfate proteoglycans (HSPGs) Syndecan (Sdc) and Dally-like (Dlp) are the only known ligands for *Drosophila* Lar (*Fox and Zinn, 2005*; *Johnson et al., 2006*). Lar interacts directly with heparan sulfate, as do mammalian Type IIa RPTPs (*Aricescu et al., 2002*). Sdc and Dlp are involved in Lar's regulation of embryonic axon guidance and larval NMJ development, but *Sdc* and *Dlp* phenotypes are much weaker than *Lar* phenotypes (*Fox and Zinn, 2005*; *Johnson et al., 2004*; *Johnson et al., 2006*), indicating that other ligands must also participate. HSPGs are not involved in R7 photoreceptor axon targeting (*Hofmeyer and Treisman, 2009*).

Mouse mutants lacking each of the Type IIa RPTPs have complex phenotypes affecting neurogenesis, axon guidance, synaptogenesis, and behavior (reviewed for PTPRD by *Uhl and Martinez, 2019*). Loss of these RPTPs also produces many phenotypes outside the nervous system. Human Type IIa RPTP polymorphisms are associated with cancer and other diseases. Analysis of Type IIa RPTP function is complicated by redundancy among the three proteins, which are very similar to each other and are expressed in overlapping patterns. Redundancy among RPTPs has been well characterized in *Drosophila*, where *Lar Ptp69D*, *Ptp10D Ptp69D*, and *Ptp10D Ptp4E* double mutants have unique phenotypes that are not present in single mutants (*Desai et al., 1997*; *Hakeda-Suzuki et al., 2017*; *Jeon et al., 2008*; *Jeon and Zinn, 2009*; *Sun et al., 2000*).

In differentiated neurons, mammalian Type IIa RPTPs are thought to function in presynaptic terminals. They can act as "synaptic organizers", facilitating formation of synapses in cell culture models via their interactions with a diverse set of postsynaptic ligands. These include Netrin-G ligand 3 (NGL-3), Tropomyosin kinase C (TrkC), Interleukin-1 receptor accessory protein-like 1 (IL1RAPL1), Interleukin-1 receptor accessory protein (IL-1RAcP), Slit- and Trk-like family protein (Slitrk) 1-Slitrk6, synaptic adhesion-like molecule (SALM) 3, and SALM5 (reviewed by *Fukai and Yoshida, 2021*; *Takahashi and Craig, 2013*). None of these ligands has a *Drosophila* ortholog, although there are *Drosophila* proteins with similar domain structures.

It has been suggested that Type IIa RPTPs function in a similar manner to neurexins, which are presynaptic proteins that interact with neuroligins and other postsynaptic ligands to induce synapse formation and/or determine synaptic properties (*Südhof, 2017*; *Takahashi and Craig, 2013*). Type IIa RPTPs and neurexins were recently shown to directly interact with each other (*Han et al., 2020a*). Two recent studies addressed the synaptic functions of Type IIa RPTPs by examining hippocampal neurons from knockout (KO) animals. One study found that cultured hippocampal triple mutant neurons lacking all three Type IIa RPTPs form synapses in a normal manner, although there was a reduction in miniature excitatory postsynaptic current (mEPSC) frequency. The only major change observed in CA3→CA1 synapses was a reduction in NMDAR-mediated EPSCs (*Sclip and Südhof, 2020*). The other study examined a conditional *PTPRS* KO, and found a decrease in the number of excitatory synapses, as well as a reduction in mEPSC frequency, but did not observe a change in AMPA vs. NMDA-receptor mediated EPSCs (*Han et al., 2020b*). The reasons for these discrepancies are unclear; it is possible that two of the Type IIa RPTPs act in opposition to each other, as has been observed in *Drosophila* for *Lar* and *Ptp99A* (*Desai et al., 1997*). Both studies examined only hippocampal synapses, so it remains possible that Type IIa RPTPs are essential for synapse formation or function in other parts of the brain.

Here we identify the IgSF CAM Sticks and Stones (Sns) as a new Lar ligand and show that the Lar-Sns interaction is conserved between flies and mammals. Sns has orthologs in *C. elegans* (SYG-2) and mammals (Nephrin). The human gene encoding Nephrin, NPHS1, is mutated in congenital nephrotic syndrome, a lethal kidney disease. Nephrin is the core component of the extracellular kidney slit diaphragm filtration network (reviewed in *Martin and Jones, 2018*).

Sns, SYG-2, and Nephrin belong to an IgSF subfamily called Irre cell recognition module (IRM) proteins, which has four members in *Drosophila*: Sns, Kirre, Roughest (Rst), and Hibris (Hbs) (*Fischbach et al., 2009*). The Sns and Hbs ECDs contain nine IgSF domains and a single FNIII repeat, while the Kirre and Rst ECDs contain five IgSF domains. Sns and Hbs are paralogs that bind to Kirre and Rst (*Bour et al., 2000*; *Galletta et al., 2004*; *Ozkan et al., 2013*; *Shelton et al., 2009*). The *C. elegans* orthologs of Sns and Kirre, SYG-2 and SYG-1, also bind to each other (*Ozkan et al., 2014*). In humans

and mice, there is one Sns/Hbs ortholog, Nephrin, and three Kirre/Rst orthologs (Nephs or Kirrels). Nephrins and Nephs interact heterophilically and homophilically. IRM proteins in different species have almost identical structures (*Ozkan et al., 2014*), and SYG-2 and SYG-1 in *C. elegans* can be replaced in vivo by human Nephrin and Neph (*Hartleben et al., 2008*; *Wanner et al., 2011*).

All four *Drosophila* IRM proteins function together as ligand-receptor pairs on the surface of founder cells and fusion competent myoblasts to regulate myoblast fusion (*Bour et al., 2000*; *Shelton et al., 2009*). The four proteins also function together in nephrocyte development (*Zhuang et al., 2009*) and in ommatidium patterning in the retina (*Bao et al., 2010*). *C. elegans* SYG-2 and SYG-1 regulate the formation of synapses by HSNL neurons onto vulval muscles. SYG-1 acts presynaptically, while SYG-2 acts in guidepost epithelial cells to direct presynaptic component assembly at the site of their interaction (*Shen, 2004*).

Mouse Nephrin and Neph1 are required for the formation and function of the kidney slit diaphragm (reviewed by *Martin and Jones, 2018*), and Nephrin is also involved in myoblast fusion (*Sohn et al., 2009*). Nephrin and the Nephs are expressed in the developing and adult nervous system (*Putaala et al., 2001*). Neph2/Kirrel2 and Neph3/Kirrel3 are involved in sorting of olfactory receptor neuron axons (*Serizawa et al., 2006*), but Nephrin's functions in the nervous system are unknown. However, since Sns and Nephrin are both involved in the development of the excretory system and in muscle fusion, it is reasonable to speculate that these orthologs might also have similar roles during nervous system development.

In this article, we show that Sns is a ligand that controls Lar's functions in R7 photoreceptor axon targeting and MB lobe development. Lar and Sns interact in *trans* in these systems because they are expressed in different sets of neurons. Sns is also required for Lar function in larval NMJ development, but in that system the two proteins are expressed in the same neurons and likely interact in *cis*.

## Results

### Identification of Sns as a Lar binding partner

Cell-surface protein (CSP) interactions mediated by ECDs are often of low affinity, having $K_d$s in the micromolar range and fast dissociation rates. Our group developed the live-dissected embryo staining screen as a way to identify low-affinity binding partners for neural CSPs expressed in a normal cellular context. It takes maximum advantage of avidity effects and has revealed low-affinity interactions that were not detected in the global in vitro 'interactome' screen, which used an ELISA-like method (the Extracellular Interactome Assay (ECIA)) to assess interactions among 200 *Drosophila* CAMs (*Bali et al., 2019*; *Bali and Zinn, 2019*; *Fox and Zinn, 2005*; *Lee et al., 2013*; *Ozkan et al., 2013*). Multimeric ECD fusion proteins are incubated with live-dissected embryos, allowing complexes of fusion proteins with CSPs on muscles or neurons to coalesce ('cap') into dense patches. The embryos are then washed directly with paraformaldehyde, which crosslinks the patches and freezes complexes into place, and the complexes are visualized with fluorescent secondary antibody (*Fox and Zinn, 2005*; *Lee et al., 2013*).

We first used this method to identify Sdc as a Lar ligand using a deficiency (*Df*) screen (*Fox and Zinn, 2005*). We later developed a gain-of-function (GOF) version of the screen in which we crossed 300 lines bearing 'EP-like' (UAS-containing) *P* elements upstream of CSP genes to a strong pancellular driver, tubulin (Tub)-GAL4, and stained the embryonic progeny of the cross with RPTP fusion proteins. We identified Stranded at Second (Sas), a large CSP expressed in epidermal cells, as a ligand for Ptp10D (*Lee et al., 2013*). To conduct the GOF screen with Lar ECD fusion proteins, we used a mutant protein, Lar$^{HS2}$, that does not bind to HSPGs (*Fox and Zinn, 2005*). We stained live-dissected stage 16 embryos from crosses of 300 EP lines to Tub-GAL4 with a dimeric Lar$^{HS2}$-alkaline phosphatase (henceforth called Lar-AP) fusion protein.

Lar-AP faintly stains central nervous system (CNS) axons in wild-type (WT) embryos. However, it brightly stained both the CNS axon ladder and the periphery in embryos from a cross between Tub-GAL4 and a line that has an insertion of an EP-like element ~200 bp 5′ to the transcription start of *sns* (*Figure 1A and B*). Quantification of Lar-AP staining intensity in the periphery showed more than a 10-fold increase in Lar-AP staining in Tub>Sns embryos vs. WT (*Figure 1—figure supplement 1*). This result shows that ectopic expression of Sns in neural, ectodermal, and muscle cells confers binding to the Lar ECD. However, it does not prove that Lar and Sns bind directly to each other since such results

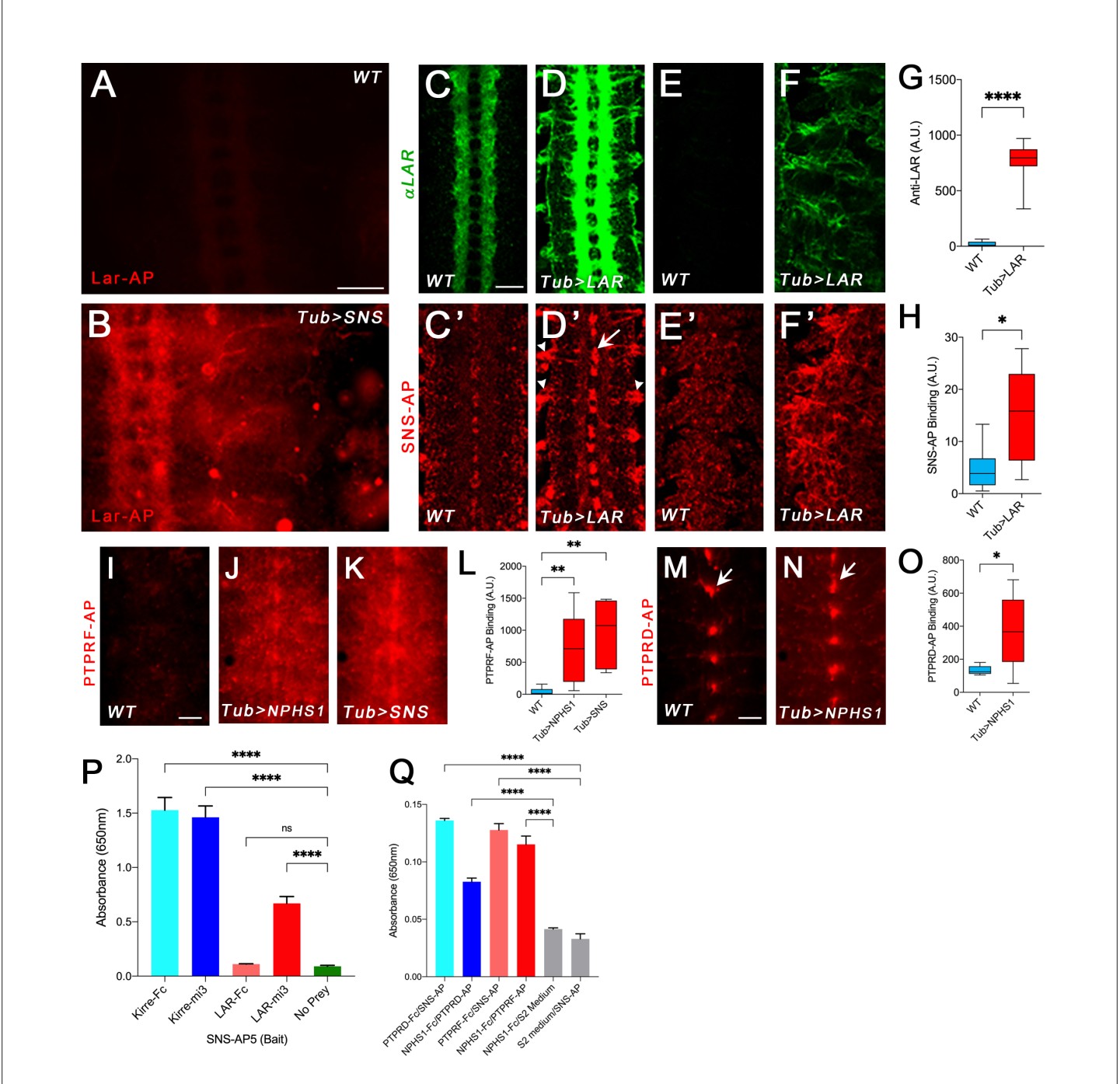

**Figure 1.** Binding of Lar and its orthologs to Sns and Nephrin. All images show live-dissected late-stage 16 embryos. (**A, B**) Staining with a version of Lar-AP (HS2) that cannot bind to heparan sulfate proteoglycans (HSPGs), visualized with anti-AP antibody. (**A**) WT embryo; Lar-AP binds weakly to central nervous system (CNS) axons (see ***Fox and Zinn, 2005***). (**B**) Tub>Sns embryo at the same exposure, showing bright ectopic staining by HS2-AP in the CNS and periphery. (**C–F**) Lar overexpression in Tub>Lar embryos, visualized with anti-Lar mAb. (**C, D**) CNS axon staining in WT (**C**) and Tub>Lar (**D**). Longitudinal axons are stained in WT; all axons are brightly stained in Tub>Lar. (**E, F**) Staining in the periphery in WT (**E**) and Tub>Lar (**F**). There is no visible staining in WT, while Tub>Lar embryos show widespread staining. (**G**) Quantitation of CNS staining with anti-Lar in WT and Tub>Lar. (**C'–F'**) Staining with Sns-AP$_5$ visualized with anti-AP antibody. (**C', D'**) CNS staining in WT (**C'**) and Tub>Lar (**D'**). Midline glia are weakly stained in WT (**C'**); note that this pattern does not resemble anti-Lar staining (**C**). Midline glia (arrow) and exit junctions (arrowheads) are brightly stained in Tub>Lar (**D'**); note the similarity between the exit junction patterns visualized with anti-Lar (**D**) and Sns-AP. (**E', F'**) Staining in the periphery in WT (**E'**) and Tub>Lar (**F'**). Staining in the periphery is increased in intensity in Tub>Lar. (**H**) Quantitation of CNS staining with Sns-AP in WT and Tub>Lar. (**I–K**) CNS staining with PTPRF-AP$_5$ in WT (**I**), Tub>NPHS1 (**J**), and Tub>Sns (**K**) embryos. Note that there is very little staining in WT, but bright staining in the entire CNS

*Figure 1 continued on next page*

*Figure 1 continued*

in Tub>NPHS1 and Tub>Sns. (**L**) Quantitation of CNS staining in WT, Tub>NPHS1, and Tub>Sns. (**M, N**) CNS staining with PTPRD-AP$_5$ in WT (**M**) and Tub>NPHS1 (**N**). Note midline glial staining in WT; this staining is only slightly increased in intensity in Tub>NPHS1 (arrows). Staining intensity in the remainder of the CNS is increased by several fold, however. (**O**) Quantitation of CNS staining in WT and Tub>NPHS1. (**P, Q**) In vitro binding measured with the ECIA assay using either AP or HRP enzymatic activity for detection. (**P**) 60-mer Lar prey (Lar-mi3) binds to Sns-AP$_5$ bait. Kirre-Fc and Kirre-mi3 preys bind to Sns-AP$_5$ equally. (**Q**) Both PTPRD-AP$_5$ and PTPRF-AP$_5$ preys bind to Nephrin-Fc (NPHS1-Fc) bait. PTPRD-AP$_5$ and PTPRF-AP$_5$ also bind to Sns-Fc bait. There is no signal with Nephrin-Fc bait and S2 medium prey or Sns-AP$_5$ prey and S2 medium bait. Scale bar, 20 μm.

The online version of this article includes the following figure supplement(s) for figure 1:

**Figure supplement 1.** Lar-AP binding quantification and ECIA assay between Hbs, Kirre, and Lar.

could also be obtained if Sns ectopic expression induced expression or stabilization of another protein that actually binds to Lar.

To address this issue, we performed 'reverse-binding' experiments to determine whether Lar binds directly to Sns. To do this, we used Tub-GAL4 to drive pancellular expression of Lar in embryos using a UAS-Lar line (*Figure 1C–F*). We then stained WT and Tub>Lar embryos with a pentameric Sns-AP$_5$ fusion protein containing the Sns ECD fused to a COMP pentamerization domain and AP (*Ozkan et al., 2013*). Sns-AP$_5$ stained the CNS in WT embryos (*Figure 1C'*). Interestingly, midline glia (arrow) were more strongly stained than axons and cell bodies. This pattern does not resemble Lar antibody staining (*Figure 1C*), indicating that Sns has another binding partner in the embryonic CNS, perhaps Kirre or Rst. There is also weak Sns-AP$_5$ staining in the periphery (*Figure 1E'*).

Embryos with ectopic expression of Lar driven by Tub-GAL4 showed a three-fold increase in Sns-AP$_5$ staining in the CNS compared to WT control embryos (*Figure 1D' and H*). Sns-AP$_5$ staining was increased at sites where motor axons exit the ventral nerve cord (VNC; arrowhead, *Figure 1D'*) and in midline glia. Staining was also increased in the periphery (*Figure 1E' and F'*). Some of this staining colocalizes with ectopic Lar, such as at the VNC exit points. This reverse binding experiment provides evidence that Lar and Sns bind to each other. We do not know why CNS longitudinal tracts, which stain brightly with anti-Lar, do not exhibit strong staining with Sns-AP$_5$. Perhaps there are access issues due to glial sheathing or there may be proteins expressed on glia and motor axons that facilitate Lar-Sns binding.

To confirm direct binding between Lar and Sns, we conducted ECIA experiments with multimerized Lar-Fc and Sns-AP$_5$ proteins made in human Expi293 cells. To obtain stronger binding to Sns in the ECIA assay, we increased avidity by making 60-mer Lar particles using the mi3 nanoparticle as a scaffold (*Bruun et al., 2018*; see Materials and methods). 60-mer Lar 'prey' exhibited a seven-fold increase in binding to Sns-AP$_5$ 'bait' coupled to the surface of an ELISA plate, showing that the two proteins interact directly in vitro. The binding signal is about twofold weaker than that obtained for the strong binding partners Sns and Kirre, which bind to each other with a K$_d$ of 2.5 μM (*Ozkan et al., 2014*). However, 60-mer Kirre prey and Kirre-Fc work equally well for detection of this strong interaction (*Figure 1P*; *Ozkan et al., 2013*; *Ozkan et al., 2014*). We also tested whether Hbs and Kirre, the other members of the IRM protein family, also bind to Lar. No binding was observed for either Hbs-AP$_5$ or Kirre-AP$_5$, while Kirre-Fc and Hbs-AP$_5$ showed strong binding, as previously observed (*Figure 1—figure supplement 1*; *Ozkan et al., 2014*).

## Mammalian orthologs of Lar and Sns bind to each other in embryos and in vitro

To determine whether binding between Lar and Sns is evolutionarily conserved, we tested whether Nephrin, the mammalian Sns ortholog, binds to PTPRD, PTPRF, or PTPRS in live-dissected embryos. To do this, we made a transgenic line with a UAS-linked full-length human Nephrin cDNA (NPHS1) construct and expressed AP$_5$ fusion proteins containing the ECDs of PTPRD, PTPRF, and PTPRS in *Drosophila* Schneider 2 (S2) cells.

We tested the binding patterns of the three AP fusion proteins in WT, Tub>NPHS1, and Tub>Sns embryos. PTPRF-AP$_5$ showed almost no staining of WT embryos, but stained the VNC and midline glia in Tub>NPHS1 and Tub>Sns embryos (*Figure 1J and K*). Staining intensity was increased by 18- to 20-fold relative to WT when Nephrin or Sns were expressed (*Figure 1L*).

PTPRD-AP$_5$ produced a clear signal in WT, with strong staining in midline glia and weak staining in the rest of the VNC (*Figure 1M*). When Nephrin was ectopically expressed using Tub-GAL4

(Tub>NPHS1), midline glial staining was slightly increased in intensity, and there was a threefold increase in staining relative to WT in the VNC as a whole (*Figure 1M–O*). These data suggest that a PTPRD binding partner is expressed in WT midline glia or that midline glial membranes bind nonspecifically to this probe. PTPRS-AP$_5$ showed little staining in WT or Tub>NPHS1 embryos.

We then tested PTPRF and PTPRD for binding to Sns and Nephrin in vitro. Fc dimers for both human proteins bound to fly Sns-AP$_5$ (*Figure 1Q*). The signal was weaker than for 60-mer Lar prey, but stronger than for Lar-Fc (*Figure 1P*). PTPRF-AP$_5$ also bound to Nephrin-Fc, and there was a smaller (but still significant) increase over background for Nephrin and PTPRD-AP$_5$ (*Figure 1Q*). In summary, these data indicate that the Lar–Sns interaction is evolutionarily conserved for at least two of the three mammalian Lar orthologs.

## Lar and Sns are co-expressed in larval motor neurons

To characterize Lar and Sns expression, we created T2A-GAL4 lines derived from *MiMIC* insertions in coding introns of the two genes (*Diao et al., 2015*). In coding intron T2A-GAL4 lines, expression of GAL4 requires in-frame readthrough from the coding region and reports on the rate of initiation of translation from the correct ATG, so these GAL4s are translational, not just transcriptional, reporters. In the third instar larval VNC, Lar-T2A-GAL4>UAS- EGFP (Lar>GFP) expression was observed in motor neurons (large paired cells) and in a large number of interneurons (*Figure 2A and B*). Sns-T2A-GAL4>UAS-EGFP (Sns>GFP) expression was also seen in motor neurons (*Figure 2C*, *Figure 2—figure supplement 1*), as well as in a pattern of interneurons that appeared different from those expressing the Lar reporter (*Figure 2D*).

Examination of NMJs showed that Lar>GFP was expressed in both types of glutamatergic motor neurons (1b and 1s) (*Figure 2K*), but not in modulatory type II or type III motor neurons (*Figure 2E and E'*). Lar>GFP expression was stronger in 1s than in 1b motor neurons (*Figure 2G and G'*). Sns>GFP was also expressed in both 1b and 1s motor neurons with stronger expression in 1s neurons (*Figure 2H, H' and L*). In the 1b neurons, Sns reporter expression levels were higher in those that target more ventral muscles, including muscles 7/6, 13, and 12 (*Figure 2J*). Sns>GFP was also observed in type II motor neurons (*Figure 2F and F'*). No Sns>GFP or Lar>GFP expression was seen in muscles.

## Lar and Sns genetically interact to shape morphogenesis of NMJs

Previous studies have shown that NMJs require appropriate levels of Lar for proper development (*Johnson et al., 2006*; *Kaufmann et al., 2002*). Reducing Lar expression causes decreases in the number of synaptic boutons at the muscle 7/6 NMJ, as well as other NMJs. *sns* homozygotes die during embryogenesis (*Bour et al., 2000*). Thus, to examine genetic interactions between *Lar* and *sns*, we combined an *sns* null mutation with two different *Lar* null mutations to analyze NMJ phenotypes in transheterozygous (transhet) animals. *sns*$^{xb3}$ is an early stop codon mutation, and *sns*$^{xb3}$ mutant embryos lack Sns protein (*Bour et al., 2000*). We tested two different alleles of *Lar* with *sns*$^{xb3}$: *Lar*$^{13.2}$ and *Lar*$^{451}$. Both have been described as null mutations (*Clandinin et al., 2001*; *Krueger et al., 1996*). *Lar*$^{13.2}$ mutants have phenotypes at the muscle 7/6 NMJ and in the larval MB (*Johnson et al., 2006*; *Kurusu and Zinn, 2008*). *Lar*$^{451}$ mutants were characterized for R7 photoreceptor defects (*Clandinin et al., 2001*). We analyzed muscle 7/6 NMJs in *Lar*$^{13.2}$/*sns*$^{xb3}$ and *Lar*$^{451}$/*sns*$^{xb3}$ transhets using a semi-automated macro in Fiji to quantify several different parameters at the 7/6 NMJ, including total NMJ area, total NMJ length, longest branch length, number of boutons, and number of branches (*Nijhof et al., 2016*). We performed separate analyses for the 1b and 1s NMJ arbor at each NMJ.

*Lar/+* and *sns/+* heterozygote controls had no 7/6 NMJ phenotypes, but the two *Lar/sns* transhets had strong phenotypes, similar to *Lar* null animals (*Lar*$^{13.2}$/*Lar*$^{451}$) (*Figure 3A–F*). Both *Lar*$^{13.2}$/*sns*$^{xb3}$ and *Lar*$^{451}$/*sns*$^{xb3}$ transhet NMJs showed severe reduction in NMJ area, number of boutons, total NMJ length, longest branch length, and number of 1b branches (*Figure 3G–K*). There was no significant difference between the stronger *Lar*$^{451}$/*sns*$^{xb3}$ transhet and *Lar*$^{13.2}$/*Lar*$^{451}$ mutants for any of the NMJ parameters measured, indicating that Lar and Sns probably function in the same genetic pathway. The 1s NMJ on muscle 7/6 is similarly affected (*Figure 3—figure supplement 1*). Other NMJs had similar phenotypes. There was no difference in the size or shape of muscles in the transhets or the *Lar* mutants. This suggests that the Lar–Sns interaction is not required for the role of Sns in myoblast fusion during embryonic development. We confirmed the NMJ abnormalities seen in *Lar/ sns*$^{xb3}$ transhets by analyzing a *sns* deficiency (*Df*) allele, which lacks the entire *Sns* gene. We observed similar

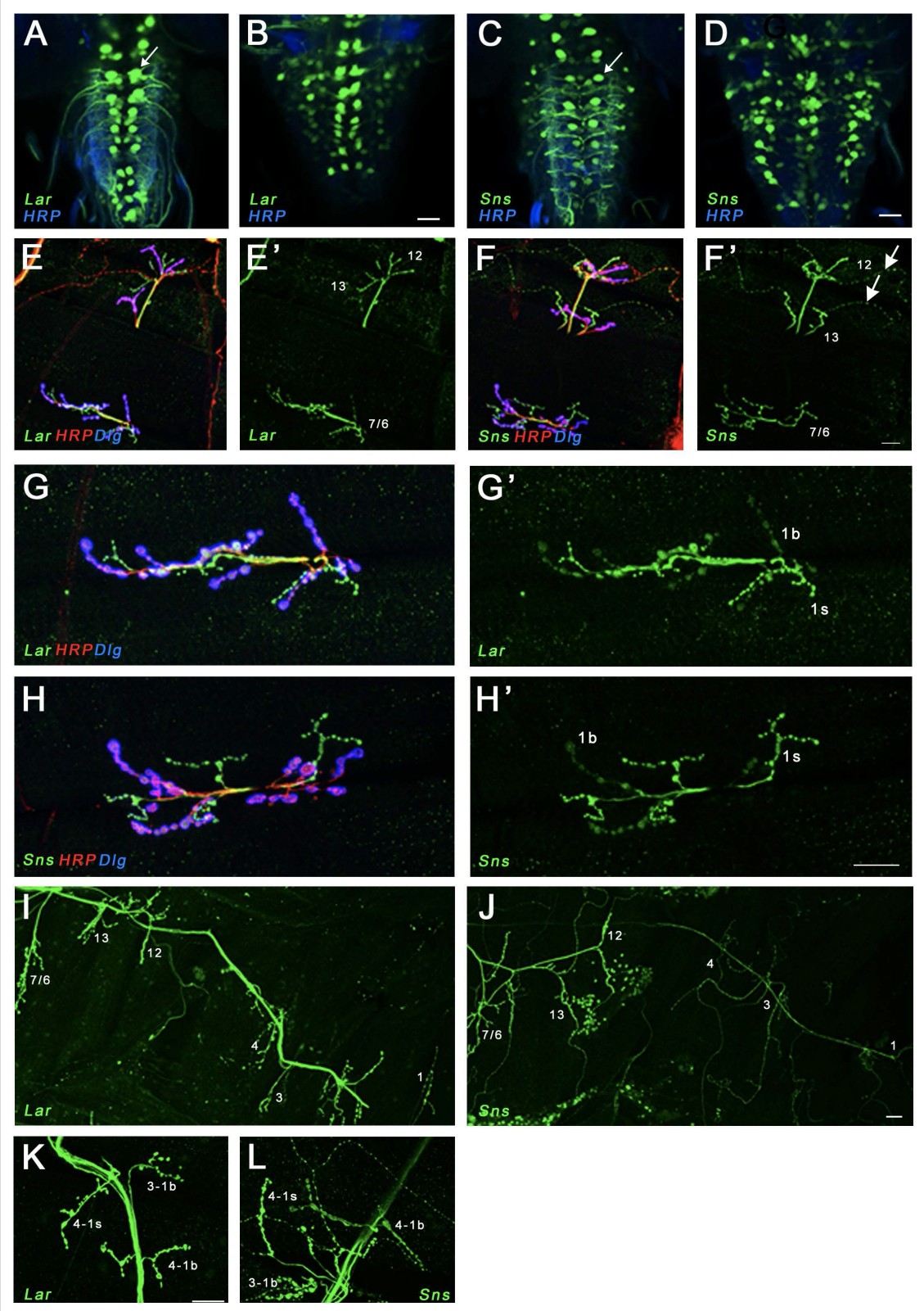

**Figure 2.** Expression of Lar and Sns reporters in motor neurons. (**A–D**) Confocal projections of 4–6 optical slices showing EGFP expression driven by either Lar^MI02154-T2A-GAL4 (Lar>GFP) or Sns^MI03001-T2A-GAL4 (Sns>GFP) (green) co-stained with anti-HRP (blue). The bright paired midline cells include motor neurons (**A, C**, arrows). (**E–H'**) Confocal projections of larval neuromuscular junctions (NMJs) on muscles 7/6, 13, and 12 (**E, F'**) and zoomed-in on muscle 7/6 (**G, H'**), triple-stained with anti-GFP (green), anti-HRP (red), and anti-Dlg (blue). (**E', F', G', H'**) show GFP signal only. Anti-HRP labels neuronal

*Figure 2 continued*

membranes, and anti-Dlg labels the subsynaptic reticulum at 1b boutons. Lar>GFP and Sns>GFP expression is seen in both 1b and 1s boutons (green), while only Sns>GFP is seen in type II boutons (**F′**, arrows). (**I, J**) Projection of optical slices through an entire larval hemisegment showing Lar>GFP (**I**) and Sns>GFP (**J**) expression in both 1b and 1s motor neurons. Individual muscles are numbered. Dorsal is to the right. Note that Lar>GFP is equally expressed in most axons and NMJs, while Sns>GFP is expressed at lower levels in axons and NMJs of motor neurons projecting to dorsal muscles. (**K, L**) Close-up of NMJs on muscles 3 and 4 showing both Lar>GFP (**K**) and Sns>GFP (**L**) expression in 1b and 1s NMJs on those muscles. Scale bar, 20 µm. See **Figure 2—figure supplement 1** for further characterization of Lar and Sns expression in the larval ventral nerve cord (VNC) and central nervous system (CNS).

The online version of this article includes the following figure supplement(s) for figure 2:

**Figure supplement 1.** Lar and Sns reporter expression in the larval ventral nerve cord (VNC) and brain.

NMJ abnormalities in *Lar^{13.2}/sns^{Df}* animals to those seen in *Lar^{13.2}/Sns^{xb3}* and *Lar^{451}/Sns^{xb3}* transhets (**Figure 3—figure supplement 2**).

We next asked whether the number of synapses was altered in *Lar/sns* transhets and *Lar* mutants at the 7/6 1b NMJ. We used antibodies against the active-zone protein Bruchpilot (Brp) to label active zones in boutons and performed quantitative analyses of Brp-positive punctae using the NMJ Fiji macro. *Lar^{451}/sns^{xb3}* transhets and *Lar^{13.2}/Lar^{451}* mutants had 64 and 71% fewer Brp punctae per 7/6 NMJ than WT (**Figure 3L and M**). This indicates that there is no compensatory increase in the number of synaptic active zones in response to reduced NMJ size and number of boutons.

## Lar and Sns act in *cis* at the NMJ

While *sns* is expressed in body wall muscles during the period of muscle fusion, its RNA levels decrease in late embryos (**Bour et al., 2000**). There was no expression of Lar or Sns reporters in muscles in third instar larvae, indicating that Lar and Sns are likely to function in motor neurons. To confirm this, we performed neuron-specific RNAi knockdown for both Lar and Sns and measured the same NMJ parameters as in the transhets and mutant analyses. We used a pan-neuronal driver, elav^{C155}-GAL4 (C155-GAL4), to drive UAS-RNAi lines for either *Lar* or *sns*. We tested two different RNAi lines for both *Lar* and *sns*. Neuronal knockdown of Lar or Sns caused NMJ abnormalities similar to those seen in *Lar/sns* transhets and *Lar* mutants (**Figure 3N–S**). The RP3 and MNISNb/d-1s axons, which form the 1b and 1s arbors of the muscle 7/6 NMJ, do not contact other motor or sensory axons after leaving the ISNb bundle, and the 1b and 1s portions of the 7/6 NMJ are separate from each other. Thus, it is likely that the 7/6 NMJ phenotypes of *Lar/sns* transhets are due to a reduction in interactions between Lar and Sns in the same neuron (in *cis*). To further confirm that Lar and Sns act in the same genetic pathway, we performed genetic experiments by reducing Sns levels in a *Lar* mutant background. If Lar and Sns act in the same genetic pathway, reducing Sns levels further in a *Lar* mutant background should not increase the severity of the Lar phenotype. *Lar* mutants (*Lar^{13.2}/Lar^{2127}*) showed reduced 1b NMJ area, number of boutons, NMJ length, longest branch length, and number of branches, as observed with other *Lar* mutants (**Figure 3—figure supplement 2**). Reducing Sns levels with neuronal *sns* RNAi in this *Lar* mutant background did not increase the severity of the 1b NMJ phenotypes (**Figure 3—figure supplement 2**), further confirming that Lar and Sns interact in the same genetic pathway. Moreover, neuronal *kirre* RNAi did not have any effect on 1b NMJs, indicating that the Sns-Kirre interaction, which plays a role in embryonic myoblast fusion and nephrocyte development, may not be involved in NMJ development (**Figure 3—figure supplement 2**).

## Lar and Sns genetically interact to control formation of the larval MB

Next, we analyzed Lar and Sns expression in the larval brain, focusing on the MB, as Lar has been shown to be required for proper development of the larval MB (**Kurusu and Zinn, 2008**). Lar was found to be expressed in Kenyon cells (KCs), the principal cells of the MB, using antibody staining. Here, we confirmed that Lar is expressed in larval KCs using Lar>GFP (**Figure 4A–B′**). A confocal *z*-projection through the entire larval MB is shown in **Figure 4B and B′**. The MB lobes are visualized using an antibody against fasciclin II (FasII), which specifically labels the MB neuropil (**Figure 4B**). A single optical slice shows that Lar>GFP labels both the dorsal (d) and the medial (m) lobes of the larval MB (**Figure 4C′**).

Sns>GFP was not detected in KCs (**Figure 4D**), but was seen in many other neurons in the larval central brain. No Sns reporter expression could be seen in either the dorsal or medial lobes of the

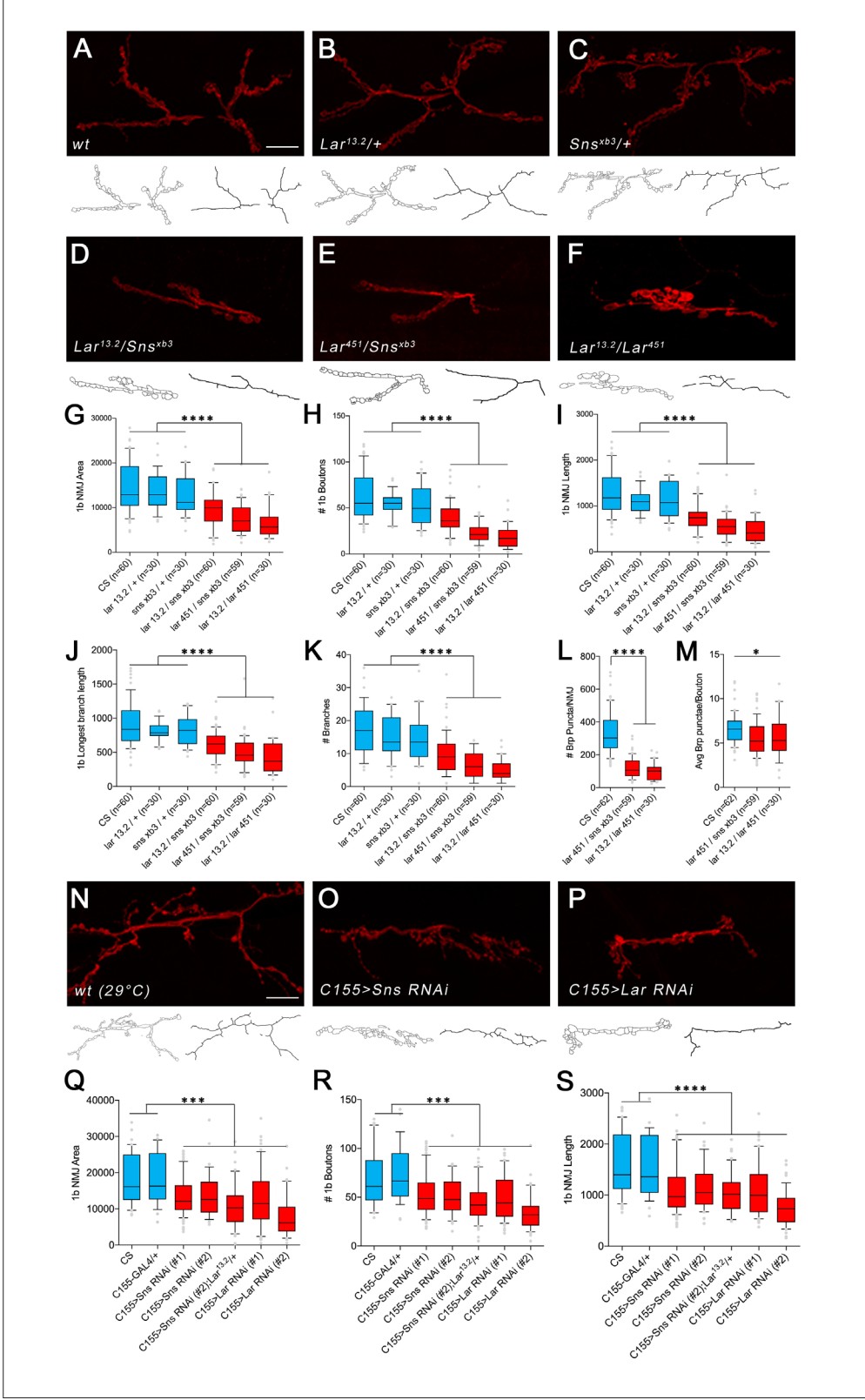

**Figure 3.** *Lar/sns* transheterozygotes have the same phenotypes as *Lar* mutants and Sns knockdowns. Neuromuscular junctions (NMJs) were analyzed using a published Fiji macro (***Nijhof et al., 2016***) that uses HRP to outline boutons and measures NMJ area, perimeter, length, longest branch length, number of branches, number of boutons, and Bruchpilot (Brp) labeled punctae. (**A–F**) Representative images of the NMJ on muscles 7/6 from

*Figure 3 continued on next page*

*Figure 3 continued*

WT and heterozygote controls (**A–C**), *Lar/sns* transheterozygotes (**D, E**), and *Lar* mutants (**F**). NMJs are labeled with anti-HRP (red). NMJ outlines showing boutons and branch architecture as outputs from the macro are under each NMJ image. (**G–K**) Quantification of 1b NMJ parameters, showing reduced NMJ size and arborization in *Lar/sns* transhets and *Lar* mutants (red) compared to het controls (blue). Data is average from segments A2–A4 from minimum 30 NMJs per genotype. (**L, M**) Quantification of Brp punctae showing reduced number of active zones in *Lar/sns* transhets and *Lar* mutants. (**N–P**) Representative images of NMJs on muscles 7/6 from animals with RNAi-mediated neuronal knockdown of Lar and Sns. Neuronal *Lar* or *sns* RNAi results in the same NMJ abnormalities seen in genetic *Lar/sns* transhets and *Lar* mutants. (**Q–S**) Quantification of NMJ parameters showing reduced 1b NMJ area, number of boutons, and NMJ length upon either Lar or Sns knockdown. A2–A4 segments were analyzed from at least 30 NMJs on muscles 7/6. All datasets were analyzed using one-way ANOVA followed by Tukey's post-hoc correction. ****p<0.0001; ***p<0.001. Scale bar, 20 µm. See *Figure 3—figure supplement 1* for analysis of 1s NMJs.

The online version of this article includes the following figure supplement(s) for figure 3:

**Figure supplement 1.** 1s neuromuscular junction (NMJ) abnormalities in *Lar/sns* transhets.

**Figure supplement 2.** 1b neuromuscular junction (NMJ) abnormalities in *Lar/sns* transhets, Sns RNAi in *Lar* mutants, and neuronal Kirre RNAi.

MB (*Figure 4E and E'*). A single optical slice of the MB lobes shows no Sns>GFP expression in the MB neuropil (*Figure 4F and F'*). In order to determine if Sns is expressed in neurons postsynaptic to MB axons, we used a dendrite-specific marker (UAS-Drep2) (*Andlauer et al., 2014*) to label Sns-expressing neurons. We observed Drep2-expressing neurons enveloping the dorsal lobes of the MB (*Figure 2—figure supplement 1*), indicating that Sns-expressing neurons are postsynaptic to Lar-expressing KCs. We also performed immunostaining using Sns>GFP and anti-Repo to label glial cells, but did not see any co-localization between GFP and Repo (*Figure 2—figure supplement 1*).

*Lar* mutants have two distinct phenotypes in the larval MB. First, the medial lobe axons fail to stop at the midline, instead crossing over to the contralateral side and forming a fused medial lobe. Second, KC axons do not branch properly (or dorsal branches do not extend after branching), resulting in reduced or absent dorsal lobes (*Kurusu and Zinn, 2008*). We investigated whether Sns is required for Lar's roles in the development of the larval MB by phenotypic analysis of the medial and dorsal lobes in *Lar/sns* transhets. FasII antibody staining specifically labels both medial and dorsal lobes. We analyzed 3D reconstructions of FasII-stained larval MBs to visualize the lobes in their entirety. Each optical section of confocal *z*-stacks through the MBs was analyzed for the medial lobe fusion phenotype. *Figure 4—figure supplement 1* shows single optical slices with medial lobe axons either intact or crossing the midline.

Heterozygous control animals did not show any abnormal phenotypes in the larval MB (*Figure 4G and H*, *Figure 4—figure supplement 1*, *Figure 4—figure supplement 2*). However, both *Lar^{13.2}/sns^{xb3}* and *Lar^{451}/sns^{xb3}* transhets as well as *Lar^{13.2}/sns^{Df}* transhets displayed fused medial lobes, similar to *Lar^{13.2}/Lar^{451}* mutants (*Figure 4I–L, M and M'*, *Figure 4—figure supplement 2*). We also observed dorsal lobe phenotypes in *Lar* mutants and *Lar/sns* transhets, with most dorsal lobes being absent in *Lar^{13.2}/Lar^{451}* and *Lar^{451}/sns^{xb3}* (*Figure 4I, K, L, N and N'*). We did not observe a strong correlation between the two phenotypes; an animal with a fused medial lobe did not always display reduced or absent dorsal lobes.

We next performed pan-neuronal RNAi knockdown for Lar and Sns using the two RNAi lines for each gene to investigate whether knocking down each gene individually also results in the MB abnormalities seen in *Lar/sns* transhets and *Lar* mutants. Similar to our genetic analyses, we found that knocking down Sns and Lar resulted in medial lobe fusion and loss of dorsal lobes. One *Lar* RNAi line (#1, HMS02186) had very strong phenotypes, while the other *Lar* RNAi line (#2, HMS00822) and the two *sns* RNAi lines had weaker phenotypes (*Figure 4—figure supplement 1*).

These data show that Lar interacts with Sns to regulate the formation of the larval MB. The Lar-Sns interaction in this context is likely to be in *trans* as we do not observe any Sns>GFP expression in larval KCs or in the MB lobes. Thus, Sns apparently functions as a Lar ligand in this system. Dendrites of Sns-expressing neurons encircle the dorsal lobes of the MB, indicating that Sns is in neurons (MBONs) that are postsynaptic to Lar-expressing KC axons (*Figure 2—figure supplement 1*). To confirm that Lar and Sns act in *trans* in the larval MB, we performed Lar and Sns RNAi specifically in MB neurons

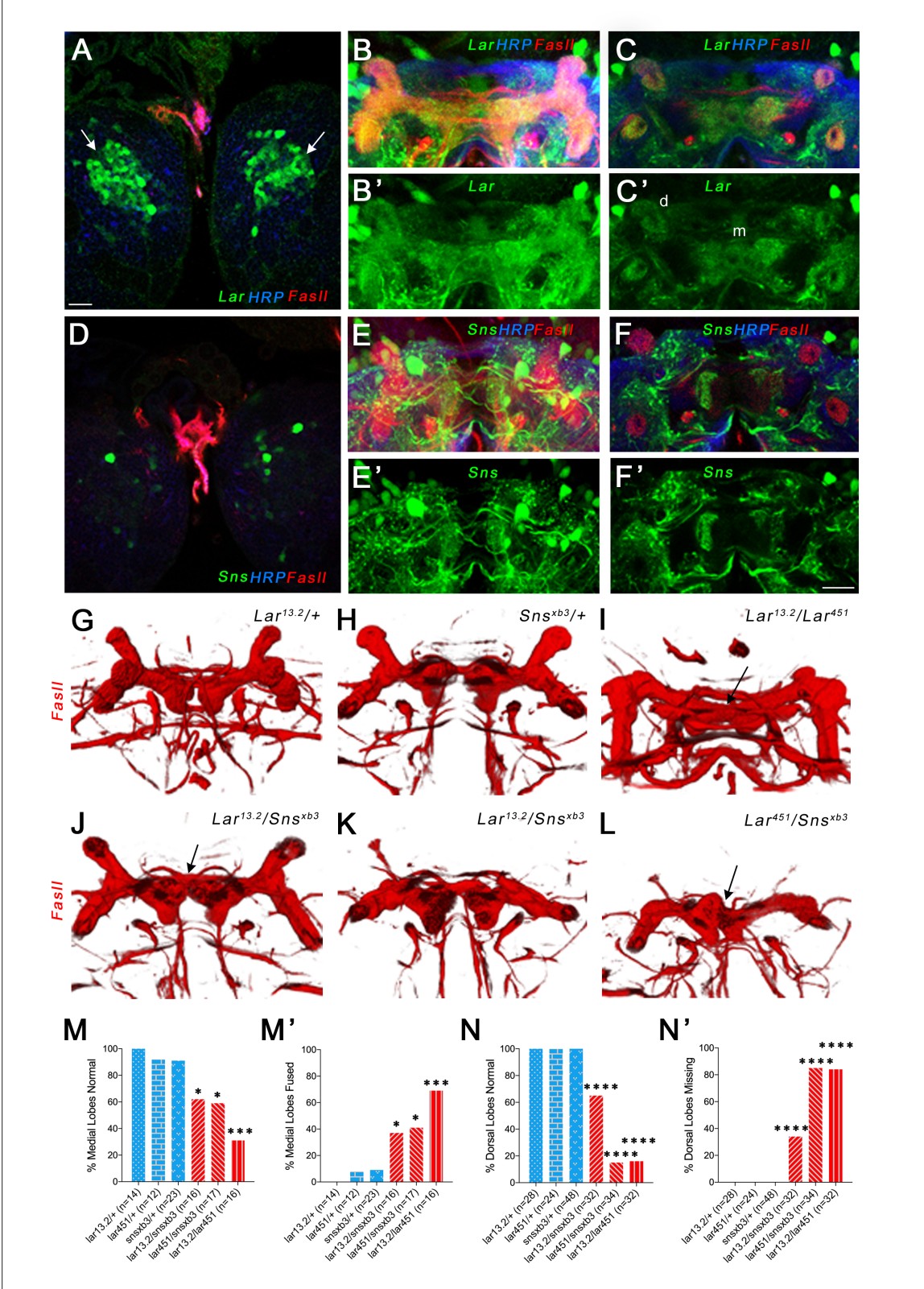

**Figure 4.** Lar and Sns act in different neurons to control mushroom body (MB) dorsal and medial lobe development. (**A–F'**) Lar>GFP and Sns>GFP expression in the larval brain. Brains were triple-stained for Lar>GFP (green), FasII (red), and anti-HRP (blue). Anti-FasII labels the MB neuropil; anti-HRP labels neuronal membranes. (**A**) Lar>GFP expression in Kenyon cells (KCs) (green, arrows). (**B, B'**) Projection of confocal slices through the entire larval MB showing Lar expression in the MB neuropil. (**C, C'**) Single optical slice showing Lar expression in the medial (m) and dorsal (d) lobes of the MB.

*Figure 4 continued on next page*

*Figure 4 continued*

(**D**) There is no Sns>GFP expression in KCs. (**E, E'**) Projection of confocal slices through the entire MB showing no overlap between Sns>GFP and the MB neuropil labeled by FasII. (**F, F'**) Single optical slice through the MB showing no Sns>GFP expression in the MB neuropil. (**G–L**) Third-instar larval MBs visualized with FasII staining. 3D reconstructions of confocal stacks using Imaris software are shown. (**G**) and (**H**) have normal MBs. (**I**) has missing dorsal lobes and medial lobe fusion (arrow). (**J**) has a medial lobe fusion phenotype (arrow). (**K**) has missing dorsal lobes. (**L**) has missing dorsal lobes and medial lobe fusion (arrow). (**M–N'**) Quantification of MB phenotypes in heterozygote controls (blue), *Lar* mutants (red), and *Lar/sns* transhets (red). In (**M**) and (**N**), the percentages of normal MBs are shown; in (**M'**) and (**N'**), the percentages of MBs with the phenotype are shown. (**M, M'**) Medial lobe fusion phenotype, (**N, N'**) Dorsal lobe branching defect. Data were analyzed using Fisher's exact test, and each genotype was compared to every other genotype. ****$p<0.0001$; ***$p<0.001$; *$p<0.05$. Scale bar, 20 µm. See *Figure 4—figure supplement 1* for single-slice analysis in *Lar/sns* transhets and Lar and Sns RNAi-mediated MB phenotypes.

The online version of this article includes the following source data and figure supplement(s) for figure 4:

**Source data 1.** Data for graphs in *Figure 4*.

**Figure supplement 1.** Medial lobe fusion in the larval mushroom body in *Lar/sns* transhets and upon RNAi-mediated Lar and Sns knockdown.

**Figure supplement 1—source data 1.** Data for graphs in *Figure 4—figure supplement 1*.

**Figure supplement 2.** Larval medial lobe fusion in *Lar13.2/SnsDf* transhets, neuronal Sns and Kirre knockdown, and mushroom body (MB)-specific Lar and Sns RNAi knockdown.

**Figure supplement 2—source data 1.** Data for graphs in *Figure 4—figure supplement 2*.

using a MB-specific GAL4 driver, OK107-GAL4, which expresses in all larval and adult MB neurons (**Aso et al., 2009**). Lar RNAi knockdown in all MB neurons resulted in medial lobe fusion in the larval MB, confirming that Lar acts in MB neurons (**Figure 4—figure supplement 2**). MB-specific Sns RNAi knockdown, however, did not cause any abnormalities in the larval MB. Combining a Lar heterozygote (*Lar13.2/+*) with MB-specific *sns* RNAi also did not cause any MB phenotypes (**Figure 4—figure supplement 2**). Combined with the Lar and Sns expression data, these data show that Lar and Sns act in *trans* in the larval MB. Neuronal *kirre* RNAi did not have any effect on the larval MB (**Figure 4—figure supplement 2**).

## Expression patterns of Lar and Sns in the pupal and adult MB

To further clarify the relationships between the Lar and Sns expression patterns, we examined Lar>GFP and Sns>GFP in the pupal and adult MB. Adult KCs are classified into three types, based on the lobes they innervate. γ neurons are born before the third-instar larval stage and form the adult γ lobe, which projects medially. α'/β' neurons are born during the late third-instar larval stage and form the α' and β' lobes, which project dorsally and medially, respectively. α/β neurons are born during early pupal stages and form the α/β lobes, which project dorsally and medially, parallel to the α' and β' lobes. α/β lobes stain with FasII antibody staining, while γ and α'/β' lobes are visualized using Trio antibody staining.

We performed immunostaining for either FasII or Trio combined with anti-GFP to label Lar>GFP and Sns>GFP in pupal and adult brains. At 24 hr after puparium formation (APF), Lar>GFP expression was detectable on growth cones of α/β KC axons near the midline in the β lobes (**Figure 5A–B'**, arrows). At 40 hr APF, higher levels of Lar>GFP expression were seen in both α/β lobes as well as α'/β' lobes. A single optical slice shows clear Lar>GFP expression in α/β and α'/β' lobes (**Figure 5F and F'**). Thus, Lar expression is high during the time period of active MB axonal outgrowth and synaptic targeting. Lar expression peaks at 72 hr APF, with strong expression in all lobes (**Figure 5—figure supplement 1A–B'**). Lar is also expressed in α/β lobes, but not in α'/β' lobes, in the adult MB (**Figure 5—figure supplement 1E–H'**).

Sns>GFP is expressed at high levels in several neuronal populations in the central brain at all three pupal stages. However, we did not observe any detectable Sns>GFP expression in either the α/β lobes or the α'/β' lobes in the 24 hr APF and 40 hr APF MB (**Figure 5C–D' and G–H'**). There is weak Sns>GFP expression in α/β lobes at 72 hr APF (**Figure 5—figure supplement 1C–D'**), but lobes with a mature morphology have already formed by this time, so this is not relevant to the lobe phenotypes we observe. Sns>GFP is also weakly expressed in adult α/β lobes (**Figure 5—figure supplement 1I–J',K–L'**).

The antennal lobes contain projection neurons (PNs) that synapse onto KCs. We observed Sns>GFP and Lar>GFP labeling of specific glomeruli at 40 hr APF and 72 hr APF (**Figure 5—figure**

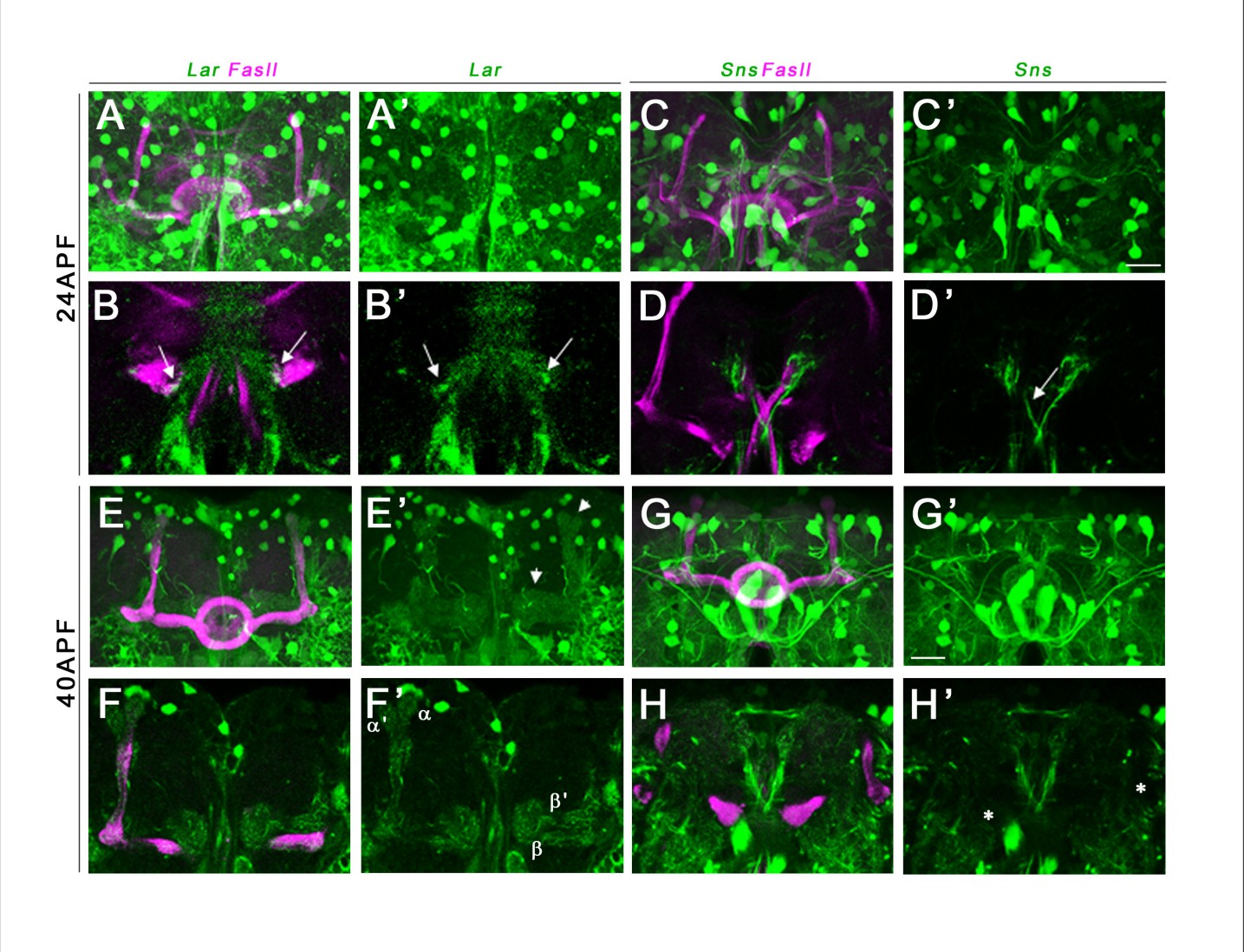

**Figure 5.** Lar and Sns expression in the developing pupal mushroom body (MB). Confocal projections and single optical slices showing Lar and Sns expression in the 24 hr after puparium formation (APF) (A–D') and 40 hr APF (E–H') pupal MB, co-stained with FasII antibody (magenta). Projections of the entire MB are shown in (A, A', C, C', E, E', G, and G'). The rest are single optical slices. Lar expression is seen in the growth cones of developing β lobe axons (B, B', arrows). No Sns expression is seen in the 24 hr APF MB (C–D'). Sns expression is seen in neuronal projections at the midline (D', arrow). (E–F') Lar expression is seen in α, α', β, and β' lobes in the 40 hr APF MB (arrows in E'; F, F', single slice showing all four lobes with Lar expression). (G–H') No Sns expression is seen in the 40 hr APF MB (asterisks in H' denote unlabeled α, α', β, and β' lobes). Scale bars, 20 µm. See *Figure 5—figure supplement 1* for Lar and Sns expression in 72 hr APF and adult brains and *Figure 5—figure supplement 2* for Lar and Sns expression in the pupal and adult antennal lobes.

The online version of this article includes the following figure supplement(s) for figure 5:

**Figure supplement 1.** Lar and Sns expression in the 72 hr after puparium formation (APF) and adult mushroom body (MB).

**Figure supplement 2.** Lar and Sns expression in the adult and developing pupal antennal lobes (ALs).

*supplement 2*). This likely represents PN expression since it has been demonstrated that Lar and Sns are enriched in PNs (*Li et al., 2020*).

## Lar and Sns genetically interact to regulate morphogenesis of α/β and α'/β' lobes of the adult MB

Having shown that Lar and Sns genetically interact to regulate the development of larval MB lobes, we then examined the adult MB to determine whether these phenotypes persist and define their

specificity for the different lobes. We used FasII immunostaining to visualize the α and β lobes and Trio to visualize the γ, α′, and β′ lobes.

Heterozygote control (*Lar*$^{13.2}$/+, *Lar*$^{451}$/+, and *sns*$^{xb3}$/+) animals all have normal α and β lobes (*Figure 6A–C and G–H′*). Note that the β lobes in these controls end well before the midline (asterisks in *Figure 6A–C*). *Lar*$^{13.2}$/*Sns*$^{xb3}$, *Lar*$^{451}$/*Sns*$^{xb3}$, and *Lar*$^{13.2}$/*Lar*$^{451}$ animals showed 75, 79, and 93% missing α lobes, respectively. *Lar/sns* transhets and *Lar* mutants also displayed midline crossing of β lobe axons, with most β lobe axons crossing the midline, creating a fused β lobe, instead of two separate lobes (*Figure 6D–H′*). The thickness of these fused β lobes was significantly greater than the normal unfused β lobes seen in control animals.

We then performed combined FasII and Trio immunostaining to visualize all the lobes of the MB. Most *Lar/sns* transhet and *Lar* null animals were missing one or more α′ lobes (*Figure 6L–N, P and P′*). β′ lobes displayed midline crossing, resulting in a single fused β′ lobe in most or all *Lar/sns* transhets and *Lar* null animals (*Figure 6L–N, O and O′*). Thus, the *Lar*, *sns*, and *Lar/sns* phenotypes were even stronger in β′ lobes than in β lobes. Some α′/β′ neurons have extended axons in third-instar larvae, and α/β axons follow the paths laid down by α′/β′ axons. Thus, the α/β and α′/β′ phenotypes observed in adults might be a consequence of guidance defects occurring in larvae. Alternatively, Lar-Sns interactions in pupae could also be instructive in guiding axons; this latter possibility is consistent with the fact that transhet phenotypes are somewhat more penetrant in adults than in larvae. In either case, the relevant Lar-Sns interaction must be in *trans* since there is no Sns>GFP expression in early or mid-pupal KCs, Interestingly, γ lobes in all animals were normal and showed no phenotype in either *Lar/sns* transhets or *Lar* nulls (*Figure 6I–N*). γ KC axons re-extend during the pupal phase, and they apparently do not require Lar or Sns for midline stopping.

To confirm that the Lar-Sns interaction is in *trans* in the adult MB, similar to the larval MB, we performed MB neuron-specific RNAi for Lar and Sns. MB-specific *Lar* RNAi resulted in β′ lobe fusion in 75% animals (*Figure 6—figure supplement 1*). *sns* RNAi in MB neurons did not cause any MB phenotypes, either alone or in combination with a *Lar* heterozygote (*Lar*$^{13.2}$/+) (*Figure 6—figure supplement 1*), suggesting that Lar and Sns act in *trans* in the adult MB as well. Pan-neuronal *sns* RNAi caused β′ lobe fusion in ~50% animals, while *kirre* RNAi had no effect (*Figure 6—figure supplement 1*).

## Lar and Sns expression in the pupal and adult OL

Lar is required in the R7 photoreceptor neuron for innervation of its target medulla layer, M6 (*Clandinin et al., 2001*; *Hakeda-Suzuki et al., 2017*; *Hofmeyer and Treisman, 2009*; *Maurel-Zaffran et al., 2001*). To analyze whether Sns is involved in this Lar function as well, we characterized expression of Lar>GFP and Sns>GFP reporters in the OL. Interestingly, although Lar has been extensively studied in the OL, there has been no characterization of Lar expression based on a GAL4 reporter. Lar antibody staining is not informative about cell-specific expression patterns because the antibody uniformly labels the neuropil and does not stain cell bodies (*Maurel-Zaffran et al., 2001*).

We observed strong Lar>GFP expression at both 40 and 72 hr APF in L1 lamina neuron cell bodies (*Figure 7C–C′ and G–G′*). Lar>GFP was also expressed at high levels in M1 and M5 layers of the medulla where L1 neurons arborize (*Figure 7D–D′ and H–H′*). It may also be present in M6, where R7 terminals are located (*Figure 7D″*), but this layer is mostly obscured by the strong staining in M5. This is consistent with results of sequencing of pupal lamina neuron mRNA, which showed that *Lar* is expressed at very high levels in pupal L1 and at low levels in R7 (*Tan et al., 2015*). We confirmed Lar expression in L1 neurons using a Lar MiMIC insertion combined with svp-GAL4>RFP, which specifically labels L1 neurons (*Figure 7—figure supplement 2*).

Lar>GFP is expressed in photoreceptor cell bodies in the 24 hr APF retina (*Figure 7—figure supplement 2B and B′*). Stronger expression was seen in R7 and/or R8 photoreceptors, which lie in the center of each ommatidium. Lar>GFP was also strongly expressed in the adult OL, with similar expression as seen in the pupal OL (*Figure 7—figure supplement 1A–B′*). There was no Lar>GFP expression in adult photoreceptors.

Many neurons expressed Sns>GFP in the 40 and 72 hr APF medullary cortex (*Figure 7E–E″ and I–I″*). Strong Sns expression was seen in both proximal and distal layers of the medulla neuropil at 40 hr APF (*Figure 7F–F″*). By 72 hr APF, Sns expression could be seen in dots at the top of the lamina, which are likely to be C2 endings (*Figure 7I′*), as well as in M1 and M5 layers of the medulla (*Figure 7J and J′*). This matches the arborization pattern of C2 neurons, which are synaptic partners

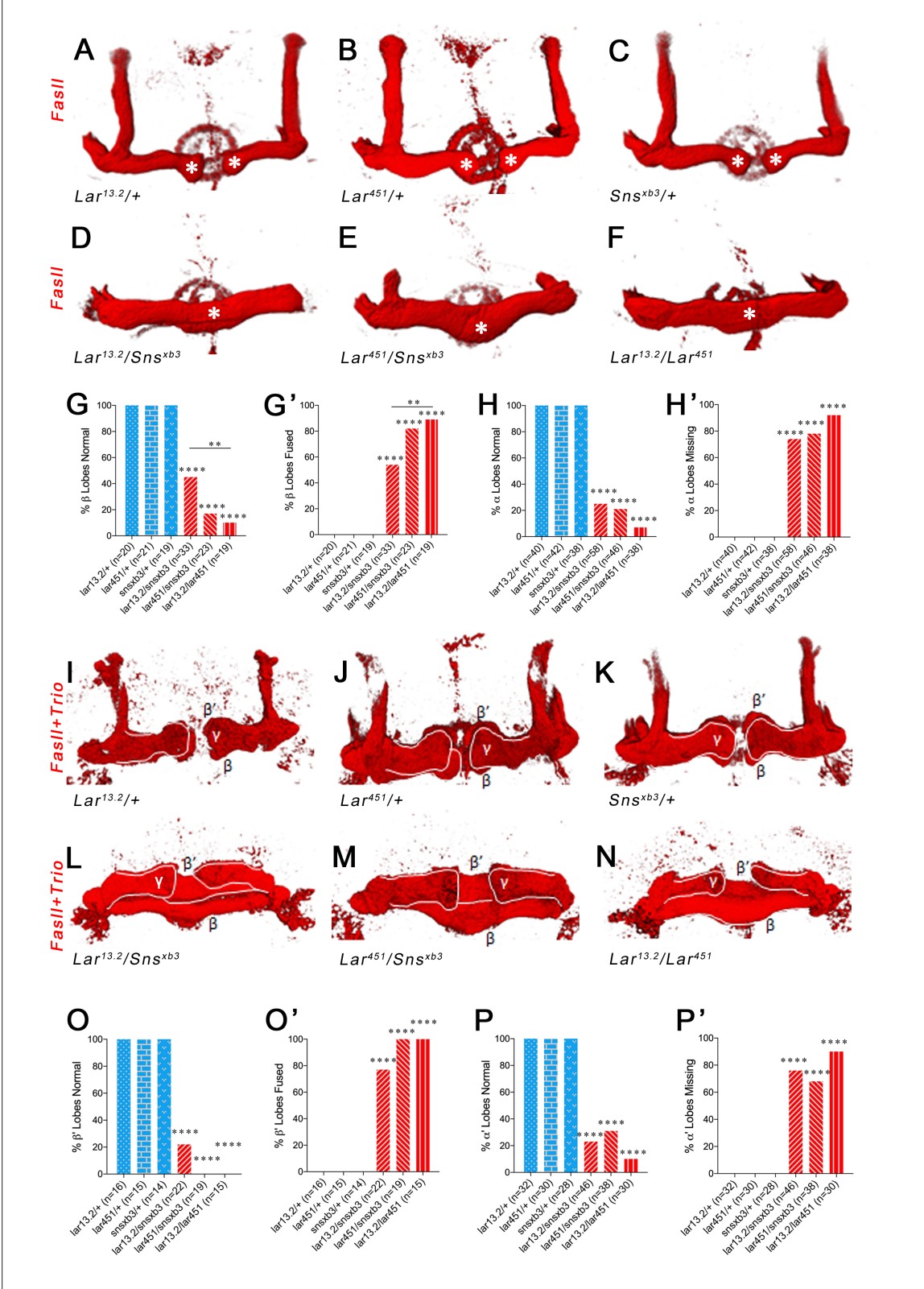

**Figure 6.** Lar and Sns are required for normal lobe development in the adult mushroom body (MB). (**A–F**) 3D reconstructions of confocal stacks from anti-FasII-stained adult brains using Imaris software. (**A–C**) Heterozygote controls showing normal α and β lobes of the adult MB. Asterisks show the ends of normal β lobes, which stop short of the midline and remain separated. (**D–F**) *Lar/sns* transheterozygotes and *Lar* mutants, showing abnormal MB architecture, with missing α lobes and β lobes fused across the midline. (**G, G′**) Quantification of β lobe midline fusion phenotype. Heterozygote

*Figure 6 continued on next page*

*Figure 6 continued*

controls (blue) show completely normal unfused β lobes. *Lar/sns* transheterozygotes and *Lar* mutants (red) have fused β lobes. (**H, H'**) Quantification of α lobe branching defect. Heterozygote controls (blue) have intact α lobes, while *Lar/sns* transhets and *Lar* mutants (red) have missing α lobes. In (**G**) and (**H**), the percentages of normal MBs are shown; in (**G'**) and (**H'**), the percentages of MBs with the phenotype are shown. (**I–N**) 3D reconstructions of confocal stacks from adult brains stained with *FasII* and *Trio* antibody to visualize the entire MB with all lobes. (**I–K**) Heterozygote controls show normal MB lobes. (**L–N**) *Lar/sns* transheterozygotes and *Lar* mutants show abnormal MB architecture, with fused β and β' lobes and missing α and α' lobes. (**O, O'**) Quantification of β' lobe midline fusion phenotype showing normal β' lobes in heterozygote controls (blue) and almost completely fused β' lobes in *Lar/sns* transheterozygotes and *Lar* mutants (red). (**P, P'**) Quantification of α' lobe branching defect. Heterozygote controls (blue) have completely normal α' lobes while *Lar/sns* transhets and *Lar* mutants (red) are missing most α' lobes. In (**O**) and (**P**), the percentages of normal MBs are shown; in (**O'**) and (**P'**), the percentages of MBs with the phenotype are shown. Data were analyzed using Fisher's exact test, and each genotype was compared to every other genotype. ****p<0.0001; **p<0.01.

The online version of this article includes the following source data and figure supplement(s) for figure 6:

**Source data 1.** Data for graphs in *Figure 6*.

**Figure supplement 1.** Adult β' lobe fusion in mushroom body (MB)-specific Lar and Sns RNAi knockdown and pan-neuronal Sns and Kirre knockdown.

**Figure supplement 1—source data 1.** Data for graphs in *Figure 6—figure supplement 1*.

of L1 neurons (*Figure 7—figure supplement 1E–F''*). C2 neurons are the only neurons with this dot-like pattern of endings, one at the top of each lamina cartridge (see *Tuthill et al., 2013*). We did not detect Sns>GFP expression in photoreceptors at any pupal stage. Sns>GFP expression persisted in numerous cells in the adult medullary cortex, with similar neuropil expression pattern as observed in the pupal OL (*Figure 7—figure supplement 1C–D'*). The pupal and adult Sns>GFP pattern appeared identical to the pattern observed with a C2-specific split-GAL4 driver (*Figure 7—figure supplement 1C–F''*).

We used Drep2 to label dendrites of Sns-T2A-GAL4-expressing neurons in the adult OL and found that the Sns expression seen in lamina and the M1 and M5 layers of the medulla at least partly represents postsynaptic elements (*Figure 7—figure supplement 2A and A'*). C2 neurons are bidirectionally connected to L1 neurons in M1 and M5 (*Takemura et al., 2013*; *Takemura et al., 2015*). Thus, Lar and Sns might interact in *trans* to regulate development of the L1-C2 circuit. Finally, we also observed strong Sns>GFP expression in the proximal layers of the medulla, as well as in the two layers of the lobula (*Figure 7—figure supplement 1C–E''*).

## Lar and Sns interact to regulate R7 photoreceptor axon targeting

In *Lar* mutants, R7 axons initially project to the correct M6 layer, but later retract to the M3 layer during mid-pupal stages (*Clandinin et al., 2001*; *Maurel-Zaffran et al., 2001*). We analyzed R7 photoreceptor axon targeting in *Lar/sns* transhets and *Lar* mutants using Chaoptin (Chp) immunostaining, which labels all photoreceptors. In the adult medulla, R7 axon endings can be clearly seen in the M6 layer by Chp (24B10) staining. In both *Lar/sns* transhets ($Lar^{13.2}/sns^{xb3}$ and $Lar^{451}/sns^{xb3}$), R7 axons failed to terminate in the appropriate M6 layer, instead retracting to the M3 layer (*Figure 8D and E*). In $Lar^{13.2}/sns^{xb3}$ transhets, 71% of R7 axons were retracted, and in $Lar^{451}/sns^{xb3}$ transhets, 63% of R7 axons were retracted (*Figure 8G*). In $Lar^{13.2}/Lar^{451}$ mutants, 88% of R7 axons retracted to M3 (*Figure 8F and G*). R7 axons that did innervate the M6 layer had abnormal terminal morphologies. Normal R7 terminals have a rounded bouton-like appearance (*Figure 8H*, arrow). In *Lar/sns* transhets and *Lar* mutants, R7 axon terminals have a spear-like appearance with thin axon terminals (*Figure 8I and J*, arrows).

Since there is no Sns>GFP expression in pupal photoreceptors, Sns must be acting in another neuronal type. M5 labeling by Sns>GFP reflects Sns expression in C2, and perhaps in other medulla neurons. Perhaps interactions between Lar in R7s and Sns in this neuron(s) facilitate R7 axon adhesion in the M6 layer and prevent retraction. Pan-neuronal Sns RNAi knockdown resulted in a small number of R7 axon retractions, as well as other R7 axon abnormalities such as incorrect innervation of neighboring R7 columns (*Figure 8—figure supplement 1*). Neuronal *kirre* RNAi did not have any effect on R7 innervation. These results suggest that Sns is the Lar ligand that controls R7 synaptic targeting.

## Discussion

We show here that Sns, a CAM known for its roles in myoblast fusion and cell patterning, is a binding partner for the Lar RPTP. We identified Sns through a GOF screen in which we crossed 300 lines

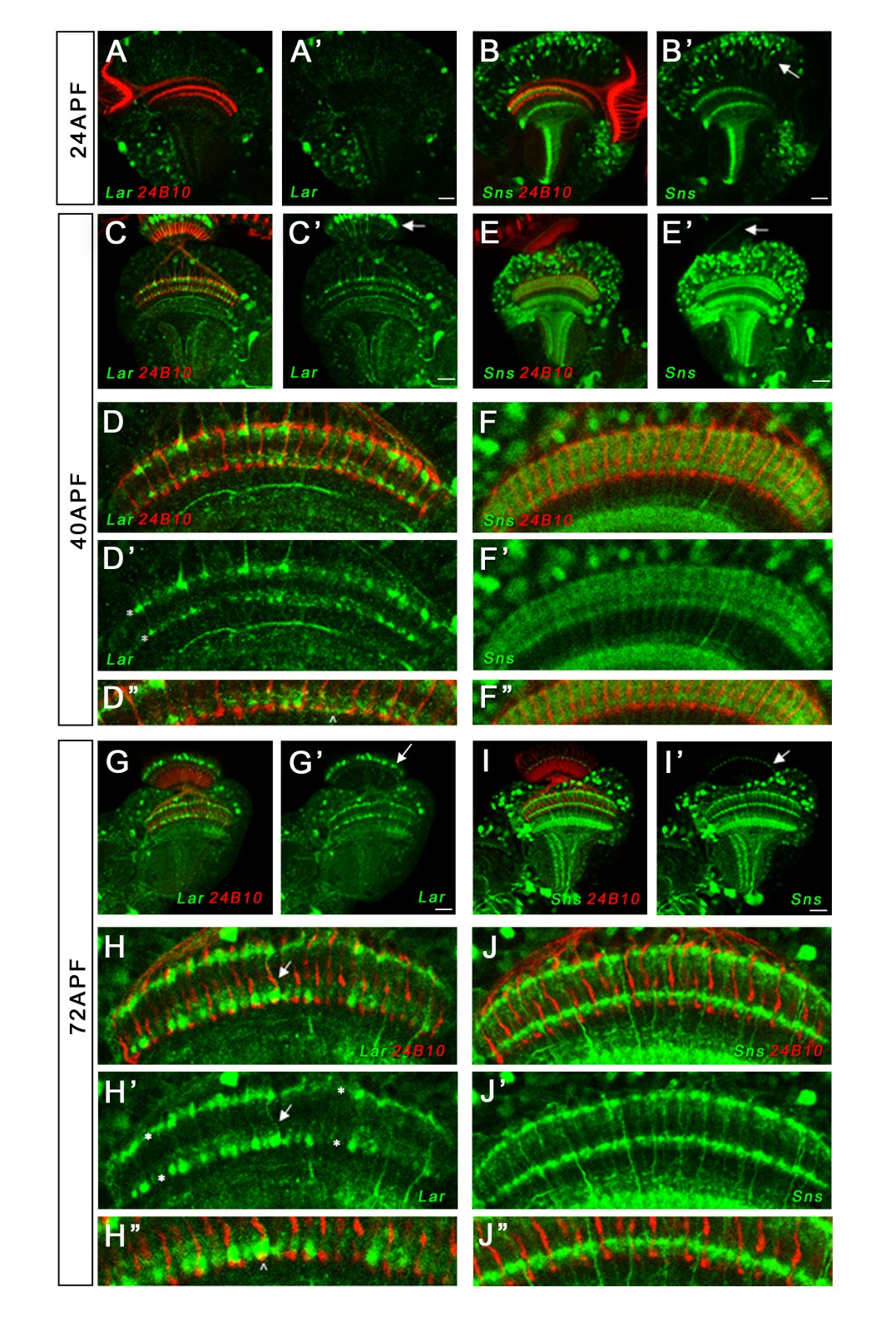

**Figure 7.** Lar and Sns expression in the developing pupal optic lobes (OLs). Single optical slices showing Lar and Sns (green) expression in 24 hr after puparium formation (APF), 40 hr APF, and 72 hr APF OL, co-stained with anti-Chaoptin (24B10 mAb, red). (**A, A'**) Weak Lar>GFP expression is seen in the medulla neuropil at 24 hr APF. (**B, B'**) At this timepoint, Sns>GFP is expressed at high levels in neuronal cell bodies in the medullary cortex (**B'**, arrow) and in specific layers in the medulla and lobula. (**C, C'**) Strong Lar>GFP expression is seen in L1 lamina neuron cell bodies (**C'**, arrow), which arborize

*Figure 7 continued on next page*

*Figure 7 continued*

in layers M1 and M5 of the medulla in the 40 hr APF OL. (**D–D"**) Close-up of the distal medulla showing L1 lamina neuron arbors in M1 and M5 layers of the medulla (**D'**, asterisks). Faint Lar expression is seen in the M6 layer of the medulla (**D"**, arrowhead). (**E, E'**) Sns>GFP expression increases at 40 hr APF, with many more neurons expressing Sns in the medullary cortex. Sns>GFP expression can be seen in several layers in the distal as well as the proximal medulla. (**F–F"**) Close-up of the distal medulla showing Sns>GFP expression in layers M1 through M5 of the medulla. Sns is not expressed in R7 photoreceptors. (**G–H"**) Strong Lar>GFP expression seen in L1 cell bodies (**G'**, arrow) and layers M1 and M5 of the medulla (**H'**, asterisks). Strong Lar expression is seen in close proximity to R7 axons and terminals (**H, H'**, arrows; **H"**, arrowhead). (**I–J"**) Sns>GFP is expressed at very high levels in the 72 hr APF OL. Specific Sns expression is seen in M1, M5, and M10 layers of the medulla and a few layers in the lobula (**I, I'**). Sns expression is also seen in the lamina. Note the dots at the top of the lamina (arrows in **E'** and **I'**), which match the morphologies of C2 endings. C2 arborizes in layers M1, M5, and M10 of the medulla. (**J–J"**) Close-up of the distal medulla showing Sns expression in M1 and M5 layers. Scale bar, 20 µm. See *Figure 7—figure supplement 1* for Lar and Sns expression in the adult OL and Sns expression in a C2-like arborization pattern. See *Figure 7—figure supplement 2* for further characterization of Lar and Sns OL expression.

The online version of this article includes the following figure supplement(s) for figure 7:

**Figure supplement 1.** Expression of Lar and Sns in the adult optic lobe (OL).

**Figure supplement 2.** Dendritic projections of Sns neurons in the adult optic lobe (OL), Lar>GFP expression in the 24 hr after puparium formation (APF) retina and Lar^MiMIC expression in L1 lamina neurons.

bearing UAS-containing *P* elements upstream of CSP genes to a pancellular GAL4 line and stained the resulting embryos with a multimeric Lar fusion protein (*Lee et al., 2013*). Lar and Sns bind directly to each other in vitro in a modified ELISA assay. This interaction is evolutionarily conserved because at least two of the three human Lar orthologs, PTPRF and PTPRD, bind to the human Sns ortholog Nephrin in embryos and in vitro (*Figure 1*, *Figure 1—figure supplement 1*).

Having shown that Lar and Sns are binding partners, we then examined whether Sns is required for Lar function in vivo by assessing the phenotypes of transheterozygous animals (transhets) lacking one copy of each gene. We also examined Lar and Sns expression to determine whether Lar and Sns are likely to interact in *cis* (on the same neuron) or in *trans* (between neurons).

*Lar* mutants have strong NMJ phenotypes (*Johnson et al., 2006*; *Kaufmann et al., 2002*), and we observed the same phenotypes in *Lar/sns* transhets and in larvae in which either Lar or Sns expression is knocked down in neurons using RNAi. Lar and Sns are both expressed in motor neurons and absent from muscles, and therefore likely work together in *cis* within a single NMJ, presumably functioning as coreceptors (*Figures 2–3*, *Figure 3—figure supplement 2*).

Lar is also required for MB development (*Kurusu and Zinn, 2008*) and is expressed in larval and pupal KCs. *Lar/sns* transhets have the same MB phenotypes as *Lar* mutants. In either genotype, dorsally projecting MB lobes fail to develop, and medially projecting lobes abnormally extend across the midline. Sns is not expressed in KCs during MB lobe development, but is expressed in neurons adjacent to the MB (*Figures 4–6*, *Figure 4—figure supplements 1 and 2*, *Figure 5—figure supplement 1*, *Figure 6—figure supplement 1*).

*Lar* acts in R7 photoreceptors to facilitate innervation of the M6 medulla layer by R7 axons (*Clandinin et al., 2001*; *Hofmeyer and Treisman, 2009*; *Maurel-Zaffran et al., 2001*). In *Lar* mutants, R7 axons initially project to M6, but then retract back to the M3 layer. We observed the same phenotype in *Lar/sns* transhets. Sns is not expressed in R7. Sns on C2 and other medulla neurons may interact in *trans* with Lar on R7 growth cones to prevent retraction of R7 terminals from the M6 layer (*Figures 7 and 8*, *Figure 7—figure supplements 1 and 2*, *Figure 8—figure supplement 1*). In summary, these data show that Sns functions as a ligand for Lar during MB and OL development. The Lar-Sns interaction is separate from the interaction of Sns with Kirre, which plays a role in myoblast fusion, nephrocyte development, and ommatidial patterning, as *kirre* RNAi does not result in the same phenotypes seen in *Lar/sns* transhets or *Lar* and *sns* RNAi (*Figure 3—figure supplement 2*, *Figure 4—figure supplement 2*, *Figure 6—figure supplement 1*).

## Lar and Sns act in *cis* to regulate NMJ morphogenesis

Lar and Sns are expressed in the same motor neurons. They are required for the development of the muscle 7/6 NMJ, as well as other NMJs. The 1b and 1s arbors at the 7/6 NMJ derive from different neurons, and both have reduced numbers of boutons in *Lar* mutants and *Lar/sns* transhets. *sns* mutants do not survive into larval stages, so we performed RNAi experiments using a neuronal driver and found that Sns and Lar knockdown resulted in the same NMJ phenotypes. The lack of Sns and

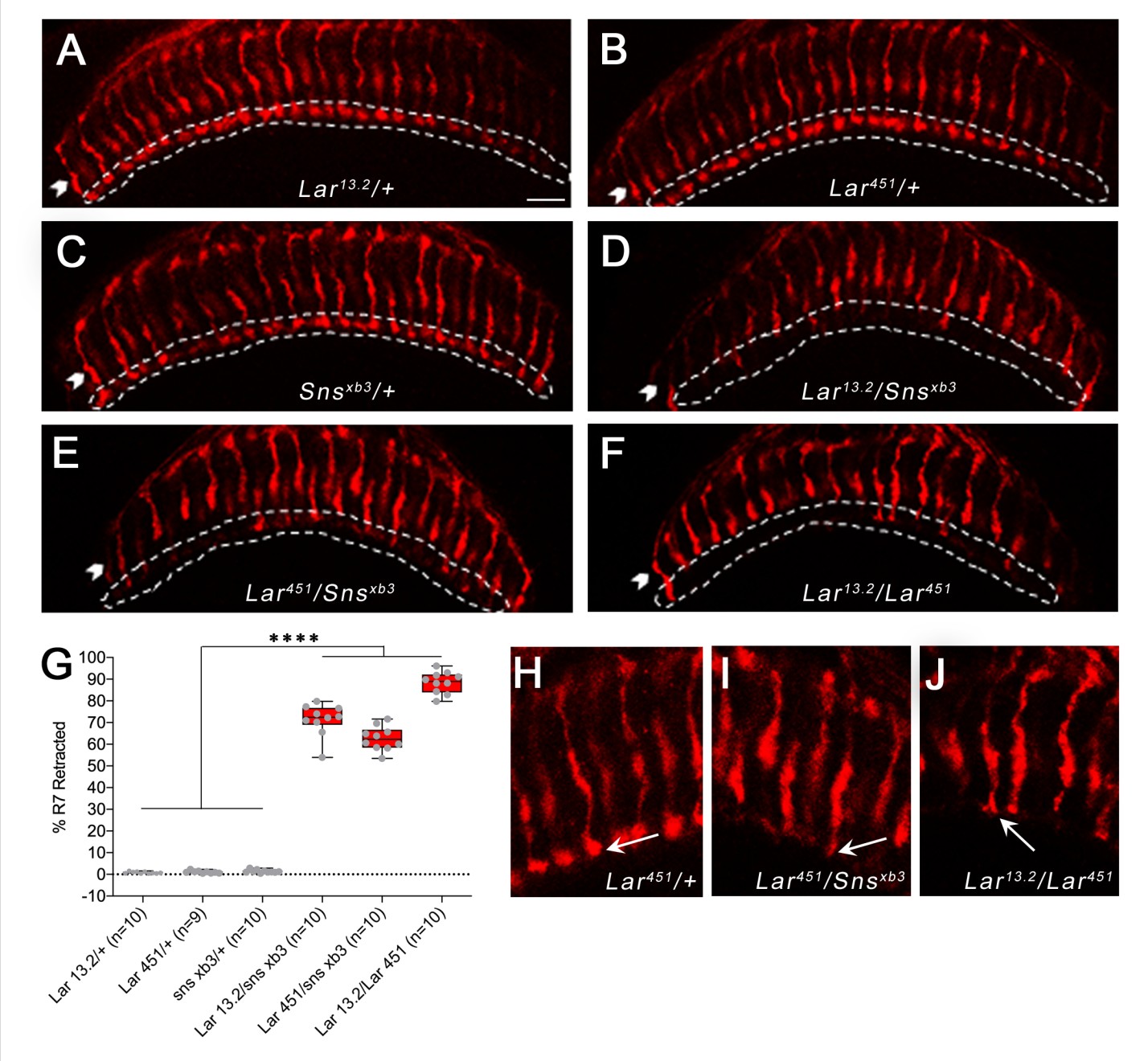

**Figure 8.** R7 photoreceptors have identical targeting defects in *Lar* mutants and *Lar/sns* transheterozygotes. (**A–F**) Single optical slices of adult optic lobes (OLs) showing R7 and R8 photoreceptors labeled for Chaoptin (24B10, red). R7 photoreceptor axons end in the M6 layer of the medulla (outlined in white), while R8 axons end in M3 layer (arrowheads). (**A–C**) Heterozygote controls showing normal R7 targeting in the M6 layer. (**D–F**) *Lar/sns* transheterozygotes and *Lar* mutants, showing abnormal R7 targeting, with most R7 axons retracting to the M3 layer. (**G**) Quantification of R7 axon retractions in control and mutant animals. R7 axons were counted in at least 10 optical slices per OL. Each data point is the average of 10–12 optical slices per OL. The data were analyzed using one-way ANOVA followed by Tukey's post-hoc correction. ****p<0.0001. (**H–J**) Single optical slices showing the morphology of R7 terminals in *Lar⁴⁵¹/+* control, *Lar⁴⁵¹/sns^xb3* transheterozygotes, and *Lar* mutants. Control animals show normal rounded bouton-shaped R7 terminals (**H**, arrow). Some R7 terminals that do not retract and stay in the M6 layer have abnormal R7 terminal morphologies, with thin and spear-shaped terminals (**I, J**, arrows). Scale bar, 20 µm.

The online version of this article includes the following figure supplement(s) for figure 8:

**Figure supplement 1.** Abnormal R7 projection examples upon pan-neuronal Sns RNAi knockdown.

Lar expression in muscles, combined with their selective expression in the motor neurons that innervate the 7/6 and other NMJs, indicates that Lar and Sns likely act in *cis* in motor neurons to regulate NMJ development (*Figures 2 and 3*, *Figure 3—figure supplement 1*). Moreover, Lar and Sns likely act in the same genetic pathway as removal of Sns in a *Lar* mutant background does not increase the severity of the *Lar* phenotype (*Figure 3—figure supplement 2*).

Lar and Sns might regulate NMJ development by forming a complex on the neuronal cell surface that regulates actin polymerization through control of tyrosine phosphorylation of Sns and other targets. The cytoplasmic domain of Sns is phosphorylated on many tyrosine residues in S2 cells (*Kocherlakota et al., 2008*). Sns could thus be a direct target for dephosphorylation by Lar. Also, two known Lar substrates, the Abl tyrosine kinase and Ena, are regulators of actin assembly (reviewed by *Johnson and Van Vactor, 2003*).

SYG-2 and SYG-1, the *C. elegans* orthologs of Sns and its binding partner Kirre, control presynaptic assembly and branch formation at HSN synapses by affecting F-actin assembly (*Chia et al., 2014*). During myoblast fusion, Sns interacts with several proteins involved with actin dynamics, including Wiskott-Aldrich syndrome protein (WASP), Solitary (Sltr)/dWIP, and the GTPase Rac. Sns signaling reorganizes the actin cytoskeleton, resulting in fusion between founder cells and fusion competent myoblasts (*Sens et al., 2010*). At *Drosophila* NMJs, bouton assembly and terminal branching are dependent upon F-actin assembly (*Koch et al., 2014*). We hypothesize that the Lar-Sns interaction may also regulate F-actin dynamics at the developing NMJ.

## *Trans* interactions between Lar and Sns regulate axon guidance of MB KCs

Our earlier work showed that *Lar* mutants display abnormal KC axon guidance in larvae, resulting in two distinct phenotypes: first, KC axons fail to stop at the midline and instead extend into the contralateral lobe, resulting in a single fused lobe. Second, KC axons fail to branch and/or extend into the dorsal lobe, resulting in a reduced or absent lobe (*Kurusu and Zinn, 2008*). Here, we show that Sns acts together with Lar to regulate axonal midline stopping and branching in both the larval and pupal/adult MB (*Figures 4–6*, *Figure 2—figure supplement 1*, *Figure 4—figure supplements 1–2*, *Figure 5—figure supplement 1*, *Figure 6—figure supplement 1*). *Lar/sns* transhets and Sns knock-down animals have the same larval phenotypes as *Lar* mutants. In the adult MB, both the α/β and α'/β' lobes are strongly affected in *Lar/sns* transhets and *Lar* mutants.

The Lar>GFP reporter is expressed in larval and pupal KCs (*Figures 4 and 5*, *Figure 5—figure supplement 1*). The Sns reporter is not expressed in larval KCs (*Figure 4*, *Figure 2—figure supplement 1*). During the pupal phase, the Sns reporter is not expressed in α'/β' KCs at any time and exhibits weak α/β KC expression only after lobe development has taken place (*Figure 5*, *Figure 5—figure supplement 1*). MB-specific RNAi for *Lar* and *sns* showed that Lar acts in KC neurons, while Sns does not (*Figure 4—figure supplement 2*, *Figure 6—figure supplement 1*). These data show that Sns is likely to interact in *trans* with Lar in the MB system. Sns is expressed in larval neurons (MBONs) that have postsynaptic elements apposed to KC axons (*Figure 2—figure supplement 1*). Perhaps Sns expressed on these and/or other neurons contacted by KC axons acts as a guidance cue.

Lar binding to Sns may generate a stop signal for medial and β' axons that prevents them from crossing the midline. β axons may cross the midline in *Lar* mutants or *Lar/sns* transhets because they follow β' axons or they may independently require Sns for midline stopping. Pupal γ axons do not rely on Lar and Sns for midline stopping.

Interactions of Lar with Sns mediate a different response in dorsal lobe and α' axons. In the absence of Lar and/or Sns, these axons fail to extend normally, and the larval dorsal and adult α/α' lobes do not form. Neurons expressing Sns may act as guideposts that are required for normal extension of these axons. The α and α' axons might respond separately to these cues or the α axons may exhibit phenotypes because they follow the α' axons. Since some α'/β' neurons are already present in larvae, adult MB phenotypes may derive from defects that occur during larval development.

## Sns is required in *trans* for Lar's roles in R7 photoreceptor axon targeting

Lar is required for R7 photoreceptor axon targeting to the M6 layer of the medulla. In *Lar* mutants, most R7 axons extend to the correct target layer, but then retract, so that the final position of the R7

terminal is usually in the M3 layer (*Clandinin et al., 2001*; *Hakeda-Suzuki et al., 2017*; *Hofmeyer and Treisman, 2009*; *Maurel-Zaffran et al., 2001*). Lar acts cell-autonomously in R7 photoreceptors to regulate targeting (*Maurel-Zaffran et al., 2001*). Consistent with this, our Lar>GFP reporter is expressed in photoreceptors during early pupal stages (*Figure 7—figure supplement 2*).

The ligand(s) responsible for Lar's actions in R7 axon targeting have not been identified. The IgSF domains of Lar are not required for R7 targeting (*Hofmeyer and Treisman, 2009*). Syndecan binds to the IgSF domains (*Fox and Zinn, 2005*; *Johnson et al., 2006*), so it can be ruled out as the Lar ligand responsible for mediating R7 targeting. Furthermore, *Sdc* mutants have no R7 retraction phenotypes (*Rawson et al., 2005*).

More than 70% of R7 terminals in *Lar/sns* transhets show retraction to the M3 layer (*Figure 8*). We also observed a small number of R7 axon abnormalities upon pan-neuronal Sns RNAi knockdown (*Figure 8—figure supplement 1*). We did not detect Sns>GFP reporter expression in R7, and RNA sequencing studies showed that R7 does not express *sns* mRNA (*Davis et al., 2020*; *Tan et al., 2015*). Hence, Lar in R7s is likely to bind to Sns in *trans*, and our data suggest that Sns is the Lar ligand that controls R7 targeting. Sns-expressing neurons arborize in layer M5, but not in M6 (*Figure 7*, *Figure 7—figure supplement 1*). Sns expression in neurons projecting to M5 may provide cues that R7 terminals employ to remain in the M6 layer. R7 terminals are bulbous structures that begin in M5 and end in M6. Thus, Sns in M5-projecting neurons might adhere to R7 terminals and prevent R7 axons from retracting. C2 neurons, which project to layers M1 and M5, appear to express Sns>GFP (*Figure 7*, *Figure 7—figure supplement 1*). C2 could be the neuron, or one of the neurons, whose expression of Sns prevents R7 axons from retracting. C2 and R7 do not form synapses with each other.

C2 receives more synapses from the L1 lamina neuron than from any other neuron and also makes many synapses onto L1 (*Takemura et al., 2013*; *Takemura et al., 2015*). *Lar* mRNA is expressed at some level in most neurons. However, RNA sequencing studies have shown that L1 neurons express *Lar* at particularly high levels (*Davis et al., 2020*; *Tan et al., 2015*), and the Lar>GFP reporter strongly labels L1 (*Figure 7*, *Figure 7—figure supplements 1 and 2*). It is attractive to speculate that Lar-Sns interactions might be important for the formation of L1-C2 synaptic connections. *kirre* and *rst* are also expressed in L1 neurons (*Kurmangaliyev et al., 2020*). Sns in C2 could thus bind in *trans* to both Lar and Kirre/Rst in L1 neurons. *kirre* RNAi in all neurons did not result in any R7 axon phenotypes, suggesting that the Sns-Kirre interaction is not important for R7 innervation.

## Evolutionary conservation of the Lar–Sns interaction

Sns is a highly conserved CAM that has orthologs in *C. elegans* and mammals (*Shen, 2004*). The *C. elegans* ortholog of Sns, SYG-2, is expressed on guidepost epithelial cells, and its interactions with SYG-1 on the HSNL neuron initiate the process of synapse formation (*Shen et al., 2004*). In *SYG-2* and *SYG-1* mutants, components of the presynaptic active zone assembly do not localize properly. This phenotype is similar to that seen in *SYD-2* mutants. SYD-2 is the *C. elegans* ortholog of Liprin-α, which binds to Lar's cytoplasmic domain. SYG-1/SYG-2 interactions recruit SYD-2 to the site of active zone assembly (*Dai et al., 2006*; *Patel et al., 2006*), and SYD-2 is mislocalized in mutants for PTP-3, the *C. elegans* Lar ortholog (*Ackley et al., 2005*). These observations suggest a model of synapse formation in which SYG-1, SYG-2, PTP-3, and SYD-2 all act together at the site of synapse assembly.

The three mammalian Type IIa RPTPs (PTPRF, PTPRD, PTPRS) bind to an overlapping set of post-synaptic binding partners that are described in the 'Introduction' section. Each of these ligands localizes to the postsynaptic membrane, where they form heterophilic *trans* complexes with presynaptic RPTPs (reviewed by *Fukai and Yoshida, 2021*; *Takahashi and Craig, 2013*). There is no evidence that Nephrin has a similar localization pattern.

Expression of a Nephrin-*lacZ* reporter was observed in many areas of the embryonic spinal cord and brain, and expression persists into adulthood in the dentate gyrus and olfactory bulb (*Putaala et al., 2001*). Phenotypes caused by loss of Nephrin in the nervous system have not been defined since standard KOs are lethal due to kidney failure and conditional KOs have not been published. We

speculate that Nephrin is a ligand for Type IIa RPTPs that is required for developmental processes controlled by these signaling receptors.

# Materials and methods

## Key resources table

| Reagent type (species) or resource | Designation | Source or reference | Identifiers | Additional information |
|---|---|---|---|---|
| Genetic reagent (*Drosophila melanogaster*) | Canton S (CS) | Bloomington *Drosophila* Stock Center (BDSC) | BDSC #64349 | |
| Genetic reagent (*D. melanogaster*) | Tubulin-GAL4 | BDSC | BDSC #5138 | |
| Genetic reagent (*D. melanogaster*) | elav$^{C155}$-GAL4 | BDSC | BDSC #458 | |
| Genetic reagent (*D. melanogaster*) | OK107-GAL4 | BDSC | BDSC #854 | |
| Genetic reagent (*D. melanogaster*) | UAS-2xEGFP | BDSC | BDSC #6874 | |
| Genetic reagent (*D. melanogaster*) | UAS-mCD8-GFP | BDSC | BDSC #32185 | |
| Genetic reagent (*D. melanogaster*) | sns$^{MIMIC-MI03001}$ | BDSC | BDSC #35916 | |
| Genetic reagent (*D. melanogaster*) | lar$^{MIMIC-MI02154}$ | BDSC | BDSC #35972 | |
| Genetic reagent (*D. melanogaster*) | sns$^{EY08142}$ | BDSC | BDSC #17434 | |
| Genetic reagent (*D. melanogaster*) | Sns-T2A-GAL4 | This paper | | Lab of Dr. Kai Zinn |
| Genetic reagent (*D. melanogaster*) | Lar-T2A-GAL4 | This paper | | Lab of Dr. Kai Zinn |
| Genetic reagent (*D. melanogaster*) | UAS-sns-RNAi | Vienna *Drosophila* Resource Center (VDRC) | VDRC #109442 | |
| Genetic reagent (*D. melanogaster*) | UAS-sns-RNAi | VDRC | VDRC #877 | |
| Genetic reagent (*D. melanogaster*) | UAS-lar-RNAi | Developmental Studies Hybridoma Bank (DSHB) | DSHB #40938 | |
| Genetic reagent (*D. melanogaster*) | UAS-lar-RNAi | DSHB | DSHB #34965 | |
| Genetic reagent (*D. melanogaster*) | UAS-kirre-RNAi | DSHB | DSHB #67340 | |
| Genetic reagent (*D. melanogaster*) | sns$^{xb3}$ | PMID #10859168 | | Gift of Susan Abmyer |
| Genetic reagent (*D. melanogaster*) | sns$^{Df}$ | DSHB | DSHB #23165 | |
| Genetic reagent (*D. melanogaster*) | lar$^{13.2}$ | PMID #8598047 | DSHB #8774 | |
| Genetic reagent (*D. melanogaster*) | lar$^{451}$ | PMID #11683994 | | |
| Genetic reagent (*D. melanogaster*) | lar$^{2127}$ | PMID# 11683993 | | |
| Antibody | Rabbit anti-GFP (rabbit polyclonal) | Thermo Fisher | #A11122 | (1:500) |
| Antibody | Rabbit anti-RFP (rabbit polyclonal) | Rockland Inc. | #600-401-379 | (1:500) |

*Continued on next page*

*Continued*

| Reagent type (species) or resource | Designation | Source or reference | Identifiers | Additional information |
|---|---|---|---|---|
| Antibody | Mouse anti-Dlg (mouse monoclonal) | DSHB | #4F3 | (1:100) |
| Antibody | Mouse anti-Bruchpilot (mouse monoclonal) | DSHB | #nc82 | (1:10) |
| Antibody | Mouse anti-Chaoptin (mouse monoclonal) | DSHB | #24B10 | (1:10) |
| Antibody | Mouse anti-FasII (mouse monoclonal) | DSHB | #1D4 | (1:3) |
| Antibody | Mouse anti-Trio (mouse monoclonal) | DSHB | #9.4A | (1:20) |
| Antibody | Mouse anti-Lar (mouse monoclonal) | DSHB | #9D8 | (1:3) |
| Antibody | Mouse anti-Repo (mouse monoclonal) | DSHB | #8D12 | (1:10) |
| Antibody | Mouse anti-Evenskipped (mouse monoclonal) | DSHB | #3C10 | (1:10) |
| Antibody | Mouse anti-human IgG (Fc specific) (mouse monoclonal) | Bio-Rad | #MCA647G | (1:200) |
| Antibody | Rabbit anti-alkaline phosphatase (AP) (rabbit polyclonal) | AbD Serotec | | (1:1000) |
| Antibody | Mouse anti-AP:biotin (mouse monoclonal) | eBioscience, Thermo Fisher | #13-9870-82 | (1:200) |
| Antibody | Goat anti-HRP-Alexa 488 (goat polyclonal) | Jackson ImmunoResearch | #123-545-021 | (1:50) |
| Antibody | Goat anti-HRP-Alexa 594 (goat polyclonal) | Jackson ImmunoResearch | #123-585-021 | (1:50) |
| Antibody | Goat anti-mouse-Alexa 488 (goat polyclonal) | Thermo Fisher | #A110029 | (1:500) |
| Antibody | Goat anti-mouse-Alexa 568 (goat polyclonal) | Thermo Fisher | #A11031 | (1:500) |
| Antibody | Goat anti-rabbit-Alexa 488 (goat polyclonal) | Thermo Fisher | #A11008 | (1:500) |
| Antibody | Goat anti-rabbit-Alexa 568 (goat polyclonal) | Thermo Fisher | #A11036 | (1:500) |
| Antibody | Goat anti-mouse-AP (goat polyclonal) | Jackson ImmunoResearch | #115-055-003 | (1:200) |

*Continued on next page*

*Continued*

| Reagent type (species) or resource | Designation | Source or reference | Identifiers | Additional information |
|---|---|---|---|---|
| Peptide, recombinant protein | Streptavidin:HRP | Thermo Fisher | #N100 | (1:500) |
| Commercial assay or kit | Ultra-TMB | Thermo Fisher | #34028 | |
| Commercial assay or kit | BluePhos | Seracare | #5120-0059 | |

## *Drosophila* genetics

WT flies used were *Canton S* (CS). *Lar* and *sns* mutants have been previously described: *Lar[13.2]*, *Lar[451]*, and *sns[xb3]* (gift of Dr. Susan Abmayr). The following lines were obtained from the Bloomington Stock Center: *sns[Df]*, C155-GAL4, tubulin-GAL4, OK107-GAL4, *sns[MiMIC MI03001]*, *Lar[MiMIC02154]*, *sns[EY08142]*, UAS-Lar RNAi (TRiP.HMS02186), UAS-Lar RNAi (TRiP.HMS00822), and UAS-Kirre RNAi (TRiP.HMC05791). Sns RNAi lines were from the Vienna *Drosophila* Resource Center: UAS-Sns RNAi (KK109442) and UAS-Sns RNAi (GD877). T2A-GAL4 lines were generated as described in *Diao et al., 2015*. Briefly, flies carrying the MiMIC insertion were crossed with flies bearing the triplet 'Trojan exon' donor. The $F_1$ males from this cross carrying both genetic components were crossed to females carrying germline transgenic sources of Cre and ϕC31. The $F_2$ males from this cross that had all four genetic components were then crossed to a UAS-2xEGFP reporter line, and the resulting progeny were screened for T2A-GAL4 transformants.

## Immunohistochemistry

Live dissections of embryos were performed as described in *Lee et al., 2013*. Briefly, egg-laying chambers were set up with adult flies and grape juice plates and left in dark at room temperature to lay eggs for 4 hr. Embryos on the grape plate were incubated overnight at 18°C, followed by 2 hr at 29°C to induce GAL4 expression. Stage 16 embryos were dissected in PBS, followed by incubation with $AP_5$ fusion protein supernatants (1× or concentrated) from lepidopteran HiFive cells infected with baculovirus vectors, transfected Schneider 2 (S2) cells, or transfected mammalian Expi293 cells for 2 hr at room temperature. Embryos were then fixed in 4% paraformaldehyde for 30 min, followed by blocking in 5% normal goat serum in 0.05% PBT (1× PBS with 0.05% Triton-X100 and 0.1% BSA). Primary antibody incubation was done overnight at 4°C. Primary antibodies used were rabbit anti-AP (1:1000, AbD Serotec), mouse anti-FasII (1:3, mAb 1D4, DSHB), and mouse anti-LAR (1:3, mAb 9D8). Following washes in 0.05% PBT, embryos were incubated in secondary antibodies for 2–4 hr at room temperature. Secondary antibodies used were Alexa Fluor 568 conjugated goat anti-rabbit and Alexa Fluor 488 conjugated goat anti-mouse (1:500, Molecular Technologies). Samples were washed and mounted in Vectashield (Vector Laboratories).

Larval dissections were performed as described in *Menon et al., 2015*. Briefly, wandering third-instar larvae were dissected in PBS and fixed in 4% paraformaldehyde for 30 min. Samples were washed in 0.25% PBT (1× PBS with 0.25% Triton X-100 and 0.1% BSA) three times and incubated overnight in 0.25% PBT at 4°C. Samples were blocked in 5% normal goat serum (in 0.25% PBT) for 1–2 hr at room temperature, followed by incubation with primary antibodies overnight at 4°C. Primary antibodies used were rabbit anti-GFP (1:500, Molecular Technologies), rabbit anti-RFP (1:500, Rockland Inc), mouse anti-Repo (1:10, mAb 8D12, DSHB), mouse anti-Eve (1:10, mAb 3C10, DSHB), mouse anti-Chaoptin (1:10, mAb 24B10, DSHB), mouse anti-discs large (1:100, mAb 4F3, DSHB), and mouse anti-FasII (1:3, mAb 1D4). Following washes in 0.25% PBT, samples were incubated with secondary antibodies overnight at 4°C. Secondary antibodies used were Alexa Fluor 488 goat anti-rabbit, Alexa Fluor 568 goat anti-rabbit, Alexa Fluor 568 goat anti-mouse, Alexa Fluor 647 goat anti-mouse (1:500, Molecular Technologies), Alexa Fluor 488 conjugated goat anti-horseradish peroxidase, Alexa Fluor 568 conjugated goat anti-horseradish peroxidase, and Alexa Fluor 647 conjugated goat anti-horseradish peroxidase (1:50, Jackson ImmunoResearch). Samples were washed and mounted in Vectashield.

Pupal and adult brain dissections were performed as described in *Menon et al., 2019* and *Xu et al., 2018*. Both pupal and adult brains were dissected in PBS and fixed in 4% paraformaldehyde (in 0.25% PBT) for 30 min. Samples were washed overnight at 4°C, followed by blocking in 5% normal goat serum in 0.25% PBT for 1–2 hr at room temperature. Samples were incubated in primary antibodies

for 2 days at 4°C. Primary antibodies used were rabbit anti-GFP (1:500, Molecular Technologies), mouse anti-Chaoptin (1:10, mAb 24B10, DSHB), mouse anti-FasII (1:3, mAb 1D4), and mouse anti-Trio (1:20, mAb 9.4A, DSHB). Samples were washed in 0.25% PBT and incubated in secondary antibodies for 2 days at 4°C. Secondary antibodies used were Alexa Fluor 488 goat anti-rabbit, Alexa Fluor 568 goat anti-mouse, Alexa Fluor 647 goat anti-mouse (1:500, Molecular Technologies), and Alexa Fluor 647 conjugated goat anti-horseradish peroxidase (1:50, Jackson ImmunoResearch). Following washes, samples were incubated in Vectashield for a minimum of 24 hr before mounting.

## Molecular biology

AP$_5$ and Fc fusion proteins were prepared as described in *Lee et al., 2013* (baculovirus) or in *Ozkan et al., 2013* (S2 cells), or in *Wojtowicz et al., 2020* (Expi293 cells). S2 cell expression vectors containing the ECDs of Sns, Hbs, Kirre, and Lar linked to AP$_5$ or Fc were obtained from the Özkan collection. These ECDs were also transferred into mammalian Fc and AP$_5$ expression vectors. To make Lar-mi3 and Kirre-mi3 particles, we produced versions of Lar-Fc and Kirre-Fc with C-terminal SpyTag3 sequences. These were expressed in Expi293 cells and purified using Ni-NTA chromatography (the proteins also have His tags). Each protein was coupled to mi3-SpyCatcher (*Bruun et al., 2018*) at ratios such that most of the 60 SpyCatcher sites were coupled to a Lar-Fc-SpyTag3 chain. To make Nephrin, PTPRD, PTPRF, and PTPRS AP$_5$ and Fc fusion proteins, ECD regions of each were amplified by PCR from full-length cDNAs and moved into pCE2 and pCE14 expression vectors using Gateway Cloning. S2 cells were transfected using Effectene transfection reagent as described in *Ozkan et al., 2013*. Expression was induced using 100 mM CuSO$_4$ 24 hr after transfection, and supernatants (sups) containing fusion proteins were collected 3 days after induction. Sups were directly used at 1× concentration for ECIA assays and were concentrated 2–5× using Amicon Ultra-15 centrifugal filter units (30 kDa molecular weight cutoff) for use in embryo staining experiments.

## ECIA

Each well of Nunc MaxiSorp 96-well plate was incubated with 50 µl of mouse anti-human IgG (Fc-specific) antibody (5 µg/ml in bicarbonate coupling buffer, pH 8.4) overnight at 4°C. Wells were washed in PBST (PBS with 0.05% Tween-20) three times for 5 min each, followed by blocking in 2% BSA (in PBS) for 2 hr at room temperature. 50 µl Fc fusion proteins were added at 1× concentration for 3 hr at room temperature, followed by washes and blocking for 30 min. 50 µl AP$_5$ fusion proteins were added at 1× concentration, pre-clustered with mouse anti-human AP:biotin conjugated antibody (1:500, eBioscience) and incubated overnight at room temperature. Wells were washed in PBST, followed by incubation with streptavidin:HRP (1:500, 50 µl per well) for 30 min. Wells were washed and incubated with 1-Step Ultra TMB HRP substrate (50 µl per well, Thermo Fisher) for 30 min protected from light. The HRP reaction was stopped by adding 2 M phosphoric acid (50 µl per well), and absorbance was measured at 450 nM. For ECIA assays with mi3 particles, each well of Nunc MaxiSorp 96-well plate was incubated with 50 µl of 100 µg/ml streptavidin overnight at 4°C, followed by addition of 50 µl of AP$_5$ bait proteins (10 ng/µl) overnight at 4°C. Wells were blocked in 2% BSA (in PBS) for 2 hr at room temperature, followed by additional blocking in biotinylated BSA for 30 min and S2 cell conditioned medium for 30 min. mi3 particles or Fc fusion proteins (50 µl) were added at 10 ng/µl or 1× concentration (for S2 cell sups) and incubated overnight at room temperature. Wells were washed in PBST, followed by incubation with either mouse anti-LAR or mouse anti-human IgG (Fc-specific) antibodies. Wells were washed and incubated with goat anti-mouse: AP antibody, followed by incubation with BluePhos AP substrate reagent (50 µl) for 30 min protected from light. Absorbance was measured at 650 nM.

## Confocal imaging and image analysis

All images were captured using a Zeiss LSM710 confocal microscope with either ×20 or ×40 objectives. NMJs were analyzed using a semi-automated macro in Fiji (*Nijhof et al., 2016*). 1b and 1s boutons were separately outlined in confocal projections and separate analyses were performed on both kinds of boutons. Dlg immunostaining was used to separate 1b and 1s boutons as 1b boutons stain brightly with Dlg and 1s boutons have very weak Dlg signals. Brp punctae were also counted using the Fiji macro. For MB medial lobe, β and β' lobe phenotypes, every confocal slice was individually analyzed for FasII-positive axons crossing the midline. For dorsal, α and α' lobe phenotypes,

confocal projections of the entire MB were analyzed for the presence or absence of lobes. For R7 photoreceptor targeting phenotype, R7 terminals in M6 layer were counted in at least 10 slices per OL with each slice being 5 μm apart. The number of R7 terminals in M6 layer was divided by the total number of R7 axons seen in M3 layer and above. Images were analyzed and processed using FIiji software.

## Statistical analysis

Data were analyzed using GraphPad Prism. For all experiments with the exception of MB phenotypes, statistical analyses were performed using one-way ANOVA followed by Tukey's post-hoc correction. MB phenotypes were analyzed using Fisher's exact test, and each genotype was compared to every other genotype from the same experiment. Box and Whisker plots show 10–90 percentile whisker span. For embryo binding experiments, sample size was 8–10 embryos per genotype. For NMJ phenotypes, sample size was 30–60 NMJs per genotype. For larval MB phenotypes, sample size was 12–20 animals per genotype. For adult MB phenotypes, sample size was 20–30 animals per genotype. For OL phenotype, sample size was 10–12 OLs per genotype. Each experiment was repeated at least three times.

## Acknowledgements

We thank Michael Anaya for discussions about in vitro binding assays, Yelena Smirnova and Annie Lam for technical assistance, Violana Nesterova for figure preparation, and Kaushiki Menon and Shuwa Xu for general discussions. We thank Susan Abmayr for *sns* lines. Imaging was performed in the Biological Imaging Facility, with the support of the Caltech Beckman Institute and the Arnold and Mabel Beckman Foundation. Protein expression in baculovirus and Expi293 cells was performed at the Caltech Protein Expression Center (Jost Vielmetter, director). This work was supported by grants from the NIH to KZ, R37 NS028182 and RO1 NS096509 and by a Gordon Ross postdoctoral fellowship to NB.

## Additional information

### Funding

| Funder | Grant reference number | Author |
|---|---|---|
| National Institutes of Health | R37 NS028182 | Kai Zinn |
| National Institutes of Health | RO1 NS096509 | Kai Zinn |
| California Institute of Technology | Gordon Ross Postdoctoral Fellowship | Namrata Bali |

The funders had no role in study design, data collection and interpretation, or the decision to submit the work for publication.

### Author contributions

Namrata Bali, H.K.L. performed the embryo GOF screen. N.B. performed all other experiments., Investigation, Writing - original draft, Writing - review and editing; Hyung-Kook (Peter) Lee, Investigation; Kai Zinn, Conceptualization, Funding acquisition, Methodology, Supervision, Writing - original draft, Writing - review and editing

### Author ORCIDs

Namrata Bali (iD) http://orcid.org/0000-0002-7219-5439
Kai Zinn (iD) http://orcid.org/0000-0002-6706-5605

### Decision letter and Author response

Decision letter https://doi.org/10.7554/eLife.71469.sa1
Author response https://doi.org/10.7554/eLife.71469.sa2

## Additional files

### Supplementary files
• Transparent reporting form

### Data availability
All data generated or analysed during this study are included in the manuscript and supporting files; Source Data files have been provided for Figure 4, Figure 4 - figure supplement 1 and 2, Figure 6 and Figure 6 - Figure supplement 1.

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
