## [Editor Report]

This article claims to identify a long-sought ligand for the receptor protein tyrosine phosphatase Lar that mediates its functions in neuromuscular junction development, mushroom body development, and photoreceptor axon targeting. This would be of interest to many developmental neurobiologists.

---

## [Decision Letter]

**Decision letter after peer review:**

Thank you for submitting your article "Sticks and Stones, a conserved cell surface ligand for the Type IIa RPTP Lar, regulates neural circuit wiring in *Drosophila*" for consideration by *eLife*. Your article has been reviewed by 2 peer reviewers, and the evaluation has been overseen by a Reviewing Editor and K VijayRaghavan as the Senior Editor. The reviewers have opted to remain anonymous.

Essential revisions:

1) Much of the fabric of observations leading to the major conclusions here are based on transcriptional reporters of gene expression, with the underlying assumption that expression is equivalent to functional action. While that is often true, and correlates well with cell-type specific expression and knock-down experiments, the authors should highlight any remaining uncertainty, as the trans-heterozygous interactions do not address the site of action and RNAi can be weak. It would seem to be important to test the trans-cellular functional interaction a bit more thoroughly. For example, if Sns and Lar loss of function was combined using a LAR null het in combination with either CIS or TRANS RNAi for Sns, we would predict that the MB or Optic lobe genetic interaction would have very clear cell-type specificity.

2) The authors should test whether Lar can interact with Hbs. If so, then genetic assays should be used to look for common functions between Lar and hbs.

3) Use a second sns allele or a deficiency to confirm the transheterozygous interaction with Lar.

4) Test whether knocking down sns in all neurons or in neurons such as C2 that project to the M5 layer disrupts R7 targeting (see below too)

5) Given the many phenotypes of Sns mutants, the authors are restricted to use of partial or conditional loss of function (LOF) in this study. This raises the question of what a strong Sns looks like in key contexts where similar phenotypes are used as evidence of requisite ligand-receptor pathway function. For this reason, in one of the contexts, it would be very helpful if the authors could use MARCM technology to show what complete loss of Sns looks like in the class of neurons that appear to selectively require its function.

6) Test whether rst or kirre mutations affect NMJ size, MB development, or R7 projections.

7) A relatively easy test of whether Sns is acting as a Lar ligand would be to express sns RNAi in a Lar null mutant background. If Sns acts by binding to Lar, its knockdown should not increase the severity of the phenotype. (Of course, this only provides a genetic test of the pathway hypothesis, and could also be used to test relative tissue-specificity of the functional contribution. Strictly speaking, it will not directly test the Ligand-Receptor binding hypothesis; something that is very hard to test in vivo without mutations that disrupt binding in vitro and function in vivo.)

---

## [Author Response]

Essential revisions:1) Much of the fabric of observations leading to the major conclusions here are based on transcriptional reporters of gene expression, with the underlying assumption that expression is equivalent to functional action. While that is often true, and correlates well with cell-type specific expression and knock-down experiments, the authors should highlight any remaining uncertainty, as the trans-heterozygous interactions do not address the site of action and RNAi can be weak. It would seem to be important to test the trans-cellular functional interaction a bit more thoroughly. For example, if Sns and Lar loss of function was combined using a LAR null het in combination with either CIS or TRANS RNAi for Sns, we would predict that the MB or Optic lobe genetic interaction would have very clear cell-type specificity.

We have performed the suggested experiments and included the new data in Figure 4—figure supplement 2 and Figure 6—figure supplement 1. As suggested, we performed *cis* RNAi for Sns in combination with a Lar null het (*Lar^13.2^/+*) and analyzed phenotypes at the larval and adult mushroom body (MB). We used a mushroom body specific GAL4 driver (OK107) to drive RNAi for either Lar or Sns in all neurons of the mushroom body. We confirmed specificity of the OK107-GAL4 line by driving UAS-EGFP reporter expression driven by OK107-GAL4 in both larval and adult brains. Lar knockdown in all MB neurons resulted in medial lobe fusion in ~50% animals in the larval MB, and β’ lobe fusion in ~75% animals in the adult MB, suggesting that Lar acts cell autonomously in MB neurons, consistent with the observation that Lar is expressed in Kenyon cells. Sns knockdown specifically in MB neurons, however, did not result in any MB phenotypes, consistent with a *trans* role of Sns in the MB. A combination of a Lar null het (*Lar^13.2^/+*) and *cis* RNAi for Sns using OK107-GAL4, also did not result in any MB phenotypes. Pan-neuronal RNAi for Sns did result in ~50% β’ lobes fused across the midline in the adult MB. These new data, along with our existing data on expression patterns of Lar and Sns in the larval and adult MB, clearly show that Lar and Sns act in *trans* in the larval and adult MB, with Lar acting cell autonomously in MB neurons and Sns functioning in a non-MB neuronal population.

2) The authors should test whether Lar can interact with Hbs. If so, then genetic assays should be used to look for common functions between Lar and hbs.

We performed the suggested experiments and have included the new data in Figure 1—figure supplement 1. We tested whether Lar and Hbs, and Lar and Kirre, bind to each other using our in vitro ECIA assay. Using a LAR-Fc fusion protein, we did not detect any binding between either LAR-Fc and HBS-AP fusion proteins, or between LAR-Fc and Kirre-AP fusion proteins. We observed a three-fold increase in binding between LAR-Fc and SNS-AP fusion proteins above background. As expected, we observed strong binding between Kirre-Fc and HBS-AP fusion proteins. This strong binding interaction between Hbs and Kirre has been previously reported. Since we did not observe any in vitro binding between Lar and Hbs, we did not proceed with genetic analyses of their interactions.

3) Use a second sns allele or a deficiency to confirm the transheterozygous interaction with Lar.

We had already mentioned in the Results section of the manuscript that we had performed genetic interactions between *Lar* and *sns* using an *sns* deficiency line as well, in addition to using the reported *sns* null allele. However, as requested, we have included these transheterozygous data using the *Sns^Df^* line in the new Figure 3—figure supplement 2 and Figure 4—figure supplement 2. We observed similar phenotypes at the larval NMJ and larval MB using *Lar^13.2^/sns^Df^* transhets, as observed using *Lar^13.2^/sns^xb3^* and *Lar^451^/sns^xb3^* transhets.

4) Test whether knocking down sns in all neurons or in neurons such as C2 that project to the M5 layer disrupts R7 targeting (see below too)

As suggested, we performed Sns RNAi knockdown experiments using either pan-neuronal RNAi (elav-GAL4) or a C2-specific driver (Tuthill 2017). While we did not observe a significant R7 retraction phenotype upon pan-neuronal Sns knockdown, we did observe abnormalities in R7 innervation in the M6 layer. These included abnormal R7 innervation of neighboring columns, as well as a low number of R7 retractions. We have included examples of these phenotypes in Figure 8—figure supplement 1. However, the R7 phenotypes did not reach statistical significance. Sns knockdown specifically in C2 neurons using a C2-specific driver, did not result in any obvious R7 phenotypes. This is probably due to the fact that the C2 driver is a split-GAL4 driver that is much weaker than the strong pan-neuronal Elav-GAL4 driver. Alternatively, it might indicate that Sns acts in some other additional neuronal population in the optic lobe (OL) to generate the R7 phenotypes. Since Sns is expressed in several different OL neuronal types, it would be difficult to speculate at this time which Sns-expressing OL neuronal population is responsible for the R7 phenotypes.

5) Given the many phenotypes of Sns mutants, the authors are restricted to use of partial or conditional loss of function (LOF) in this study. This raises the question of what a strong Sns looks like in key contexts where similar phenotypes are used as evidence of requisite ligand-receptor pathway function. For this reason, in one of the contexts, it would be very helpful if the authors could use MARCM technology to show what complete loss of Sns looks like in the class of neurons that appear to selectively require its function.

We agree that it would be interesting and important to find out the effects of complete loss of Sns, we were unable to perform MARCM analyses of Sns in the MB and OL due to the fact that we do not yet know the specific neuronal populations where Sns acts in both of these contexts. We have confirmed that Sns does not act in MB neurons based on MB-specific Sns RNAi. Moreover, since Sns is not expressed in any photoreceptor neurons at any developmental stage, it is unlikely that Sns acts in these neurons in the OL. Thus, we would not expect to see any phenotypes in the MB and OL if we were to perform MARCM analyses using MB specific or photoreceptor-specific Sns clones. We already observe very strong phenotypes at the larval NMJ upon neuronal Sns knockdown. The NMJ phenotypes upon Sns knockdown are as strong as Lar knockdown. Moreover, since Sns is expressed in motor neurons and not in larval muscles, and we observe NMJ phenotypes upon neuronal knockdown of Sns, we can conclude that Sns acts in motor neurons at the larval NMJs. Thus, we do not anticipate gaining much additional insight into Sns’s actions at the NMJ by performing motor neuron-specific MARCM analysis of Sns.

6) Test whether rst or kirre mutations affect NMJ size, MB development, or R7 projections.

As suggested, we performed *kirre* RNAi analyses at the larval NMJ, larval and adult MB and adult OL. We have included these additional data in Figure 3—figure supplement 2, Figure 4—figure supplement 2 and Figure 6—figure supplement 1. We could not use a *kirre* null allele, since *kirre* null animals do not survive to third instar larval stages, similar to *sns* null animals. Neuronal knockdown of Kirre using Elav-GAL4 did not result in any abnormalities in either 1b or 1s bouton numbers, NMJ area, length or branch numbers at the 7/6 NMJ (Figure 3—figure supplement 2). Pan-neuronal Kirre knockdown also did not result in any abnormalities in either the medial lobe or the dorsal lobes of the larval MB (Figure 4—figure supplement 2). Similarly, the adult MB was also morphologically normal upon Kirre knockdown in all neurons. The α/β, α’/β’ and the γ lobes of the adult MB were not different from controls (Figure 6—figure supplement 1). We also looked at R7 projections in the adult OL upon Kirre RNAi knockdown and did not observe any obvious R7 phenotypes.

7) A relatively easy test of whether Sns is acting as a Lar ligand would be to express sns RNAi in a Lar null mutant background. If Sns acts by binding to Lar, its knockdown should not increase the severity of the phenotype. (Of course, this only provides a genetic test of the pathway hypothesis, and could also be used to test relative tissue-specificity of the functional contribution. Strictly speaking, it will not directly test the Ligand-Receptor binding hypothesis; something that is very hard to test in vivo without mutations that disrupt binding in vitro and function in vivo.)

As suggested, we have performed this experiment and included the new data in Figure 3—figure supplement 2. We performed this experiment in the context of the larval NMJ, as the *Lar* null phenotype at the NMJ is ~50% penetrant. On the other hand, the *Lar* null phenotypes at the MB and OL are ~70-90% penetrant. The high penetrance of *Lar* null phenotypes in the MB and OL would make it difficult to conclude the contribution of adding Sns RNAi in these contexts. We used a combination of two *Lar* alleles which gives a slightly lower penetrance at the larval NMJ (*Lar^13.2^/Lar^2127^*). This *Lar* null combination shows ~40-50% reduction in 1b NMJ area, number of boutons, NMJ length, longest branch length and number of branches at the 7/6 NMJ, compared to ~65% reduction seen with *Lar^13.2^/Lar^451^*. Since the NMJ phenotype seen in *Lar^13.2^/Lar^2127^* nulls is not too strong, if Sns and Lar are acting in separate pathways, we would expect the severity of this phenotype to increase upon addition of Sns RNAi. Consistent with our other observations, we did not observe an increase in the severity of the NMJ phenotype upon adding neuronal RNAi knockdown of Sns to *Lar^13.2^/Lar^2127^* mutants. *Lar^13.2^/Lar^2127^*; elav-GAL4>UAS-Sns RNAi animals also show ~40-50% reduction in all 1b NMJ parameters. These observations strongly point towards Lar and Sns acting in the same genetic pathway at the larval NMJ.